# CodeTrek: Flexible Modeling of Code using an Extensible Relational Representation

**Pardis Pashakhanloo**
University of Pennsylvania

**Aaditya Naik**
University of Pennsylvania

**Yuepeng Wang**
Simon Fraser University

**Hanjun Dai**
Google Research

**Petros Maniatis**
Google Research

**Mayur Naik**
University of Pennsylvania

## Abstract

Designing a suitable representation for code-reasoning tasks is challenging in aspects such as the kinds of program information to model, how to combine them, and how much context to consider. We propose CodeTrek, a deep learning approach that addresses these challenges by representing codebases as databases that conform to rich relational schemas. The relational representation not only allows CodeTrek to uniformly represent diverse kinds of program information, but also to leverage program-analysis queries to derive new semantic relations, which can be readily incorporated without further architectural engineering. CodeTrek embeds this relational representation using a set of walks that can traverse different relations in an unconstrained fashion, and incorporates all relevant attributes along the way. We evaluate CodeTrek on four diverse and challenging Python tasks: variable misuse, exception prediction, unused definition, and variable shadowing. CodeTrek achieves an accuracy of 91%, 63%, 98%, and 94% on these tasks respectively, and outperforms state-of-the-art neural models by 2-19% points.

## 1 Introduction

Deep learning techniques are increasingly applied to code-reasoning tasks, including bug detection (Allamanis et al., 2018), type inference (Hellendoorn et al., 2018), code summarization (Alon et al., 2019b), program repair (Dinella et al., 2020), and code generation (Alon et al., 2020), among many others. The successful application of these techniques to a given task depends heavily on the *program representation* that encompasses relevant program features and model architecture.

There are many crucial choices involved in designing a suitable representation for a new task. Consider the example in Figure 1 which depicts an instance of an exception-prediction task (Kanade et al., 2020), whose goal is to predict the exception type in the placeholder "[??]" in the highlighted statement. Predicting the correct type AssertionError from a range of 20 pre-defined exception types in Python requires understanding the implementation of the check() function, which is defined in a different class (TestObject). Thus, the desirable context for the model goes beyond the immediate lexical neighborhood of the placeholder, possibly inside a chain of called functions.

The richness of information and extended scope inevitably imply that the relevant context may be very large. Models tackling such tasks are challenged to either reduce the scope of a task—e.g., a single function, or a few contiguous lines of code text—or heuristically sample from larger scope to produce a small enough input to fit inside the memory of a GPU. For example, Transformers learn to reason about code from a sequence of tokens in the program; and GNNs with $n$ layers—a hyper-parameter which, for message-passing architectures, determines how much of the graph is *reachable* from some immediately adjacent context to the task example—prune all nodes that are further than $n$ graph hops away from the placeholder. However, distance is not always the determining factor in collecting relevant information. For instance, when trying to decide which exception type is applicable, it may be more important to follow control flow edges until they meet the raising of an exception than, say, fetching all adjacent statements without exception handling.

We outline two key design goals motivated by these considerations. First, we observe that programming languages have well-defined semantics, which ensures that relevant information (module imports such as pickle, class inheritance, inter-procedural control flow and data flow, deeper analyses

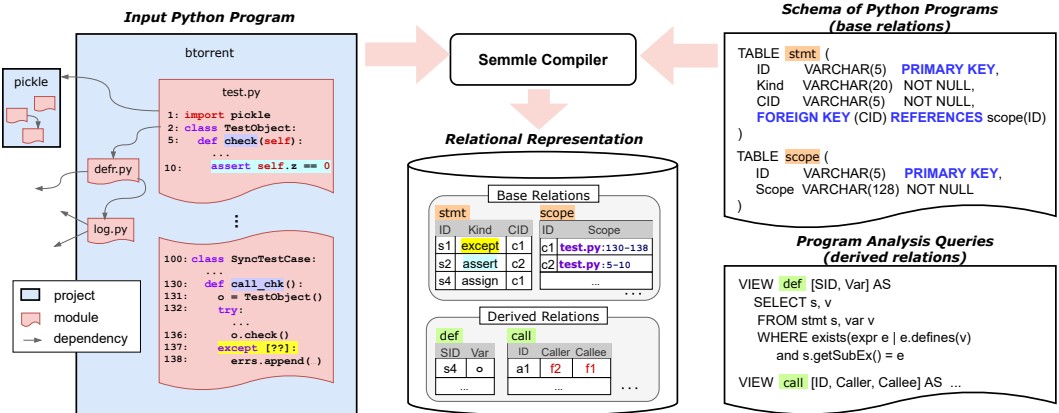

Figure 1: Example showing how CODETREK translates an exception-prediction sample in a Python program into a feature-rich representation that consists of base relations that capture the program's syntax and derived relations that capture semantic information computed by program analysis queries.

like def/use chains and object escape, etc.) can be extracted via a number of deterministic analyses. So, instead of learning this kind of information indirectly from labeled data, we can make it directly available to the model. The model can thus focus on learning the information that can only be discovered from rich contexts. Second, even when rich information is easily accessible, making well-informed predictions in code-reasoning tasks requires intelligent context collection to fit the needs of the task. So, instead of solely considering the model's technical constraints, we must capture relevant context in a task-specific manner.

In this paper, we propose CODETREK, a deep learning approach that realizes these goals. CODETREK leverages a declarative program analysis framework to produce a rich, easily extensible representation of context as a relational database, and a biased graph-walk mechanism for pruning that context in a task-specific way before presenting it to a model based on Transformers and DeepSets (Zaheer et al., 2017). CODETREK builds upon Semmle (Avgustinov et al., 2016), which converts codebases in C, Java, Python, etc., into relational databases that capture the underlying structure and semantics of code, as well as a query language, CodeQL, for specifying program analyses to compute new semantic information. CODETREK brings little modelling innovation—the architecture is reminiscent of neural techniques for knowledge-graph reasoning (Das et al., 2018; Perozzi et al., 2014). However, it is useful both as a relation generator and a task generator. As the former, it can harness existing and new analyses to add more inductive bias for hard tasks. As the latter, it can help benchmark neural techniques on more challenging tasks, and generate auxiliary training objectives to pre-train unsupervised code-understanding models (Feng et al., 2020; Guo et al., 2020; Kanade et al., 2020).

We evaluate CODETREK on four diverse tasks on real-world Python programs. They include two existing tasks, *variable misuse* and *exception prediction*, as well as two newer ones, *unused definition* and *variable shadowing*. The newer tasks are sophisticated CodeQL queries, written by program analysis experts, and enable testing the power of neural models: they both involve complex logical reasoning, and only 1.6% of the *unused definition* samples contain bugs, which is more in line with real-world settings. CODETREK achieves an accuracy of 91%, 63%, 98%, and 94% on these tasks respectively, which is 2-19% points higher than state-of-the-art neural models CuBERT, GREAT, GGNN, and Code2Seq. We also demonstrate the robustness of CODETREK in two out-of-distribution scenarios: real-world *variable misuse* samples from GitHub and *unused definition* samples involving subtle code perturbations introduced using a systematic test-generation framework, Skeletal Program Enumeration (Zhang et al., 2017). CODETREK achieves an accuracy of 57% and ROC-AUC of 78%, respectively in these scenarios, which is 6–11% points and 14–36% points higher than the baselines.

In summary, this paper makes the following contributions:

1. We propose to represent programs as relational databases that make rich context readily available for code-reasoning tasks using deep learning.
2. We present a graph-walk mechanism that prunes the unrelated context in a task-specific manner.
3. We propose techniques to enable task designers to easily tailor and stress-test their models via program analysis queries, walk specifications, and systematic test-program generation.

4. We identify two new challenging tasks for neural code reasoning, *unused definition* and *variable shadowing*; although sophisticated, non-neural static-analysis tools can solve them, these tasks pose a useful litmus test for neural code-reasoning frameworks and demonstrate the ability of CODETREK to generate hard tasks that follow real-world program distributions with modest effort.

5. We extensively evaluate our approach and demonstrate that deeper relational information about code helps neural models outperform the state-of-the-art in terms of accuracy and robustness.

CODETREK is publicly available at `https://github.com/ppashakhanloo/CodeTrek`.

## 2 THE CODETREK FRAMEWORK

### 2.1 BACKGROUND

Inspired by the idea of storing codebases as databases, CODETREK represents a program as a relational database. Specifically, CODETREK leverages the per-language schema defined by Semmle to uniformly store lexical, syntactic, and semantic program information as *base relations* in the database—we focus on Python in this paper, but the approach is language-agnostic, as long as Semmle supports the language. Each relation contains information—in the form of tuples—about a particular kind of program element, such as expressions, statements, and so on. The columns of a relation specify its attributes. For instance, in Figure 1, tuple (s1, except, c1) in the stmt relation specifies that s1 is an except statement contained in a scope with identifier c1, and tuple (s4, o) in the def relation specifies that variable o is defined in some statement with identifier s4. The schema also defines *referential integrity constraints* of the form R.A → S.B where A is called a foreign key of referencing relation R, and B is a unique attribute (e.g. a primary key) of referenced relation S. For example, in Figure 1, we have stmt.CID → scope.ID.

### 2.2 A BIRD'S-EYE VIEW OF CODETREK

Facilitated by CODETREK's uniform representation of programs, task developers can easily obtain new semantic information by writing program-analysis queries in CodeQL, an SQL-like language. The newly derived information is also in the form of *derived relations*, which maintains the uniformity of the relational representation. The derived information is stored in def, which, together with call, can bias the prediction of the best variable to replace a placeholder. A task developer need not be a machine-learning expert to bring in more semantic information about programs: All they need do is write a CodeQL query, and the resulting derived information will be added to the existing richness of the program's available features in CODETREK.

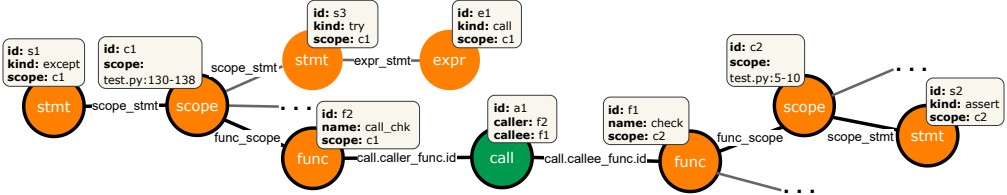

Figure 2: A partial illustration of a graph generated by CODETREK.

CODETREK translates a relational database to a graph whose nodes correspond to tuples, and whose edges follow referential integrity constraints. An example of such a graph is illustrated in Figure 2 where each node is depicted as a circle along with its type (e.g., func) in white font. Orange and green nodes correspond to tuples of base and derived relations, respectively. The attributes (e.g., name, kind, etc.) of the node are shown in a box at the corner of the node. For each referential integrity constraint R.A → S.B, an edge type R.A_S.B is defined, connecting the tuples of the two relations with the same value on the edge attributes R.A and S.B. For brevity of presentation in Figure 2, when there is a single such constraint between a pair of relations R and S, we omit the attributes from the edge type. But we do not omit them when there are multiple such constraints, such as in the case between relations func and call, namely, call.caller → func.id and call.callee → func.id. This graph view of program semantics helps extract succinct context as input to a model. Context extraction from the resulting CODETREK graph is done via biased random walks of the graph, in a fashion specified by the task definition. The starting node—which we call an *anchor* node—may be example-specific (e.g., the node containing the placeholder) or task-specific (e.g., all nodes holding a variable declaration). In Figure 2, the node that represents tuple stmt(s1, except, c1)—which corresponds to the statement on

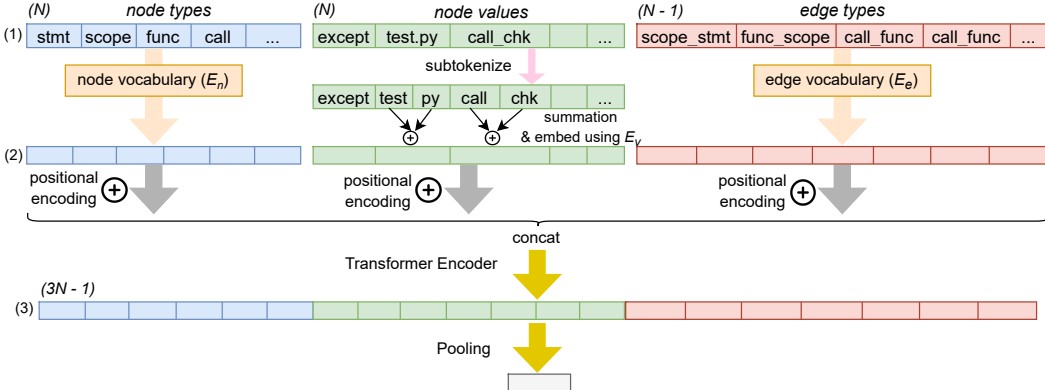

Figure 3: Embedding of the walk highlighted in Figure 2.

line 137 in Figure 1 (left)—is the anchor because the goal of the task is to predict a suitable exception type in the except statement. The walk generator traverses the graph by biasing traversal of edges according to each neighbor's node type. If no bias is specified, walks are simply fair random walks. Different probability mixes for different node types encourage the model to sample walks more relevant to a task. An example of such a walk is shown in Figure 2 using circles with thicker borders. This walk reaches the "assert" node which in fact determines the exception type that should be used in the except statement. For instance, to spend more time traversing longer-range dependencies in other functions, the developer can assign a higher value to call nodes. In our evaluation, we assign higher probabilities to nodes of types stmt and expr. We could achieve improved accuracy for the tasks compared to baselines by only modifying the probabilities of up to 4 types of nodes. Learning the walk specification given the task without human input is exciting future work.

Finally, to convert random walks to a distributed representation, CODETREK embeds each walk (including the types and attributes of each node and edge in the walk) using a Transformer encoder, and then produces an order-invariant representation of the set of walks using the Deep Set architecture (Zaheer et al., 2017). The resulting hidden representation can then be used by to make predictions for the particular code-reasoning task.

Random walks to embed a graph are perhaps a regressive choice, compared to more modern solutions such as GNNs or relational Transformers. For instance, DeepWalk (Perozzi et al., 2014) uses random walks for distributed-representation learning in a transductive setting, and Code2Seq (Alon et al., 2019a) uses shortest paths between pairs of AST leaf nodes. We chose biased random walks for several reasons. First, this enables CODETREK to choose task-specific strategies to heuristically fetch relevant context for a task, rather than choosing to embed all tokens of a function or class in lexical order. Second, in contrast to GNNs, this enables CODETREK to potentially follow much longer chains through the semantic graph than what would be possible in a message-passing GNN of a tractable number of layers—this was, in fact, the motivation behind the model architectures by Hellendoorn et al. (2020). Finally, in contrast to other walk-based approaches such as Code2Seq and AnyCodeGen, which share some of our motivation, our graph structure has much richer connectivity and is larger. For instance, those approaches only consider paths ending at token-bearing leaf nodes, with only AST interior nodes in between, whereas CODETREK admits arbitrary paths through the graph. In our example, paths with more than two token nodes, e.g., the walk illustrated in Figure 2 using circles with thicker borders is admissible for CODETREK but not for Code2Seq. CODETREK builds upon the above techniques and extends to relational graphs, with different sampling strategies and neural architectures that would suit database representation better.

### 2.3 BUILDING BLOCKS OF CODETREK

We now describe briefly the building blocks of CODETREK. Pre-processing consists of two steps: (a) turning the codebase into a relational database (*Code2Rel*), using a pre-existing set of analyses and the developer's own analyses, and (b) mapping the relational database to a graph (*Rel2Graph*). Then, a graph is processed into sets of graph walks to present to a model. Finally, we train using a cross-entropy loss to implement a particular task.

**Codebases as Relational Databases.** Code2Rel applies to the codebase the system's base program analysis queries, which make up the base relations, as well as those provided by the developer, which

form the derived relations. The result is a database comprising of a number of named tuples for each relation type. For Python, we collect 95 base relations and 277 derived relations, although adding more derived relations is simply an exercise in writing a few lines of CodeQL. Optionally, this step can limit the resulting database to only those relations that are reachable (through query-to-query dependencies) from some program analyses marked as required by the developer. Appendix D.1 provides further details.

**Constructing Relational Graphs from Databases.** Rel2Graph interprets the relations produced by Code2Rel as a graph, as follows. Each named tuple is represented by a node with the values of the tuple attributes as its features. Edges are added between these nodes as described in subsection 2.2 such that the edge type R.A_S.B is defined for each referential integrity constraint R.A → S.B between nodes representing tuples of relations R and S.

**Representing Code as a Set of Walks.** Given a code database that is converted to graph $G$ via the above Rel2Graph, we propose to represent it by the embedding of a set of walks $W$, via the procedure *Graph2Walks*. Graph2Walks projects a code graph as produced by Rel2Graph to a set of walks, according to the task-specific walk specification (anchor node predicate, traversal bias, and target walk length). Graph2Walks samples from the distribution of such random walks, by repeatedly picking a node satisfying the anchor predicate, and traversing up to a maximum number of neighbors, following the transition probabilities specified. The resulting walks are collected as token sequences of relation types of nodes, their attribute values, and the edge types traversed, in the order of traversal. Figure 3 row (1) shows such a walk representation corresponding to the walk highlighted in Figure 2. Appendix D.3 specifies Graph2Walks precisely.

**Embedding the Set of Sampled Walks.** Given a walk $w = [n_0, e_0, n_1, e_1, \ldots, n_{N-1}]$ of length $N$ steps consisting of $N$ nodes and $N-1$ edges, we produce an initial embedding $X_w \in \mathbb{R}^{(3N-1) \times d}$, where $d$ is the embedding dimension. It consists of three segments. The first $N$ rows of the embedding tensor represents the $N$ node types (relation names), using an embedding lookup in $E_n \in \mathbb{R}^{R \times d}$, where $R$ is the number of relations. The next $N$ rows represent the attribute values of the $N$ nodes; we subtokenize attribute values (using a $V$-sized WordPiece vocabulary for attribute values), embed each subtoken using $E_v \in \mathbb{R}^{V \times d}$, and mean-pool the subtoken embeddings into each node's attribute embedding. The last segment represents the $N-1$ edge types (recall that an edge type is a tuple of two relation names and primary-key/foreign-key attributes), using an embedding lookup in $E_e \in \mathbb{R}^{I \times d}$, where $I$ is the number of referential-integrity constraints in the database. All three embedding matrices $E_n, E_v, E_e$ are learnable parameters. Each individual part of the embedding tensor gets its own sinusoidal positional encoding (denoted as $\text{PE}_w \in \mathbb{R}^{(3N-1) \times d}$). We use a Transformer encoder to represent the embedding of walk $w$ as $\mathbf{e}_w = pooling\Big(\text{Transformer}(X_w + \text{PE}_w)\Big) : \mathbb{R}^{(3N-1) \times d} \mapsto \mathbb{R}^d$, where after the last layer of Transformer we do mean-pooling over all $3N-1$ elements of the walk, to obtain a $d$-dimensional $\mathbf{e}_w$. The steps for embedding of a walk sampled from the graph in Figure 2 is illustrated in Figure 3.

**Training and Inference.** An example consists of a set of walks and a ground-truth label, $(W, \hat{y})$. Given an unordered set of walk embeddings $\{\mathbf{e}_w\}_{w \in W}$, we build a classifier by using the construction $y = \text{MLP}\big(\text{DeepSet}(\{\mathbf{e}_w\}_{w \in W})\big)$, where $y$ denotes the predicted label and we optimize for cross entropy loss. However, for a binary classification task, we can obtain a more interpretable model via $y = \sum_{w \in W} \alpha_w \sigma(\text{MLP}(\mathbf{e}_w))$, where $\sigma(\cdot)$ is the sigmoid function, and $\alpha_w = \frac{\exp \text{MLP}(\mathbf{e}_w)}{\sum_{w' \in W} \exp \text{MLP}(\mathbf{e}_{w'})}$. This way, we can inspect the individual walks that contributed the most (i.e., the highest $\alpha_w$) to the positive or negative predictions, and see if that aligns with human reasoning. We refer to $\alpha_w$ as the *walk score*. We train using the Adam optimizer with 8 GPUs for distributed synchronized SGD training (see Appendix A for details).

## 3 EVALUATION

**Tasks.** We consider two main criteria in selecting tasks. The first is *locality*, which is determined by whether reasoning within a function typically suffices, or whether inter-procedural reasoning is required. The second is *declarativity*—whether the task can be stated as a logic problem that can be solved using declarative queries. For declarative tasks, we write queries in CodeQL (detailed in Appendix C); for non-declarative tasks, we rely on available datasets. We treat the following tasks:

1. VARMISUSE. Given a function and a variable accessed in it, predict whether the variable is misused. We also consider a variation of this task, VARMISUSE-FUN (Kanade et al., 2020), that

| Task | CODETREK | GGNN | Code2Seq | GREAT | CuBERT |
|------|----------|------|----------|-------|--------|
| VARMISUSE | **0.91** ± 0.003 | 0.69 ± 0.004 | – | 0.82 ± 0.002 | 0.89 ± 0.003 |
| VARMISUSE-FUN | 0.70 ± 0.004 | 0.54 ± 0.004 | 0.52 ± 0.005 | **0.89** ± 0.003 | 0.84 ± 0.003 |
| EXCEPTION | **0.63** ± 0.003 | 0.28 ± 0.02 | 0.30 ± 0.01 | 0.44 ± 0.008 | 0.42 ± 0.008 |
| EXCEPTION-FUN | 0.65 ± 0.01 | 0.51 ± 0.02 | 0.51 ± 0.008 | 0.68 ± 0.007 | **0.69** ± 0.007 |
| DEFUSE ∗ | **0.98** ± 0.002 | 0.76 ± 0.07 | – | 0.84 ± 0.05 | 0.76 ± 0.01 |
| DEFUSE-FUN ∗ | **0.91** ± 0.005 | 0.77 ± 0.07 | 0.66 ± 0.01 | 0.82 ± 0.007 | 0.71 ± 0.01 |
| VARSHADOW | **0.94** ± 0.007 | 0.71 ± 0.01 | 0.70 ± 0.01 | 0.93 ± 0.008 | 0.91 ± 0.008 |

Table 1: Accuracy results of CODETREK. Rows that are marked by ∗ are measured by ROC-AUC, and the rest are measured by accuracy. The best performance in each row is denoted in boldface.

takes only a function and predicts whether *all* variables are used correctly in the function. Note that neither variation is declarative: given a well-formed program, no logic query can deterministically decide that a variable is misused, since that decision depends on the intended semantics.

2. EXCEPTION. Given a module containing a masked exception type in an except clause, predict the most appropriate built-in exception type out of 20 choices. We also consider a variation of this task, EXCEPTION-FUN (Kanade et al., 2020), that is similar to EXCEPTION but takes a single function as scope. Although EXCEPTION needs inter-procedural reasoning, neither variation is declarative, since the choice of the appropriate exception type is subjective in Python.

3. DEFUSE. Given a function and a variable definition in its scope, predict whether the definition is used. We also consider a variation of this task, DEFUSE-FUN, that takes a function as its input and predicts whether *any* definitions are unused. This task is especially interesting because the real-world distribution of programs that contain unused definitions is skewed. Both variations are declarative (see Appendix C.1) and require only intra-procedural reasoning.

4. VARSHADOW. Given a module, predict whether any variable defined within a certain scope has the same name as a variable defined in an enclosing outer scope, thereby shadowing that latter variable. Similar to EXCEPTION, this task requires inter-procedural analysis in order to reason over both local as well as global variables. It has a declarative query (see Appendix C.2).

**Benchmark.** We use the ETH Py150 Open corpus consisting of 125K Python modules[1]. It is a de-duplicated and redistributable subset of ETH Py150[2]. Specifically, for the non-declarative tasks, we use the datasets released by Kanade et al. (2020). Since these are function-level samples but the EXCEPTION task is module-level, we augment the function in each sample with the entire containing module for this task. For the declarative tasks, we use analyses written in CodeQL to annotate all functions (or modules, as applicable) in ETH Py150 Open. All datasets consist of real examples, except for VARMISUSE-FUN and VARMISUSE where variable misuses are synthetically introduced into real code. We collected a number of apparent variable misuses from GitHub commits to test our models and baselines on a realistic dataset. Dataset details are available in Appendix H.

**Baselines.** To compare CODETREK's performance with state-of-the-art techniques, we select four baselines: we implement GGNN by Allamanis et al. (2018) and Code2Seq by Alon et al. (2019a), build classifiers on top of the GREAT encoder by Hellendoorn et al. (2020), and fine-tune the pre-trained Python model for CuBERT by Kanade et al. (2020), which is essentially the Transformer-based classifier implementation of BERT. For Code2Seq we use ASTs as base program structures as described by Alon et al. (2019a). We sample leaf-to-leaf paths from these ASTs. The number of paths we sample is the same as the number of walks we sample for training CODETREK models. For GGNN and GREAT, we compute the data flow, control flow, and lexical information described by Allamanis et al. (2018) and Hellendoorn et al. (2020), respectively, using Semmle CodeQL and augment program ASTs with those edges. We detail baseline hyperparameters in Appendix A, and fine-grained information about the size of AST-based versus CODETREK graphs in Appendix G.1.

## 3.1 ACCURACY OF CODETREK

We evaluate the performance of CODETREK and the baseline techniques on all the tasks described in section 3, all presented as classification tasks. We perform 10-fold cross-validation and report the average of the metric that we use to measure the performance of each task. We use ROC-AUC as the metric for DEFUSE and DEFUSE-FUN tasks due to their unbalanced datasets, and accuracy for the

---

[1]https://github.com/google-research-datasets/eth_py150_open
[2]https://www.sri.inf.ethz.ch/py150

| Task | CODETREK | GGNN | Code2Seq | GREAT | CuBERT |
|------|----------|------|----------|-------|--------|
| VARMISUSE-REAL | **0.57** | 0.51 | 0.50 | 0.49 | 0.46 |
| DEFUSE-SPE ∗ | **0.78** | 0.53 | 0.63 | 0.41 | 0.47 |

Table 2: Robustness results of CODETREK.

remaining tasks. The results are reported in Table 1. In 5 out of 7 tasks, CODETREK outperforms GGNN, Code2Seq, GREAT, and CuBERT by 2–19% points.

There are various reasons why CODETREK performs better than these approaches. First, declarative tasks such as DEFUSE-FUN (or DEFUSE) require complex reasoning about the interactions between program variables. For instance, one needs to reason about the uses and definitions of variables in a flow-sensitive manner to determine whether any unused definitions exist in a program. Consider the code snippet in Figure 4. The definition of the variable month on line 2 is unused but that at line 3 is used in line 4. CODETREK gives a majority of the walks sampled using the definition at line 3 a high score (around 0.99), indicating the existence of a use of that definition. However, most of the walks sampled from the definition of month at line 2 were given a lower score, and so CODETREK determines that this definition is unused. We observe that both CuBERT and GREAT fail to distinguish between the definitions on lines 2 and 3, and so they predict both to be used.

```
1  def get_month(self, t):
2    month, _, _ = t
3    def validate(month):
4      return is_valid(month)
5    return self.month
```

Figure 4: Example DEFUSE-FUN task.

Additionally, some tasks such as EXCEPTION require reasoning beyond the boundaries of a single function to make informed predictions. Functions in the chain of function calls can be lexically far from each other, thus rendering the limited context size of CuBERT and GREAT insufficient. We observe that GGNN fails in the presence of such long call chains on par with findings of Alon & Yahav (2021). CODETREK addresses this kind of mispredictions by readily using a call graph relation to connect the chain of function calls. This enables CODETREK to traverse a long distance without the need to consider other statements in the program that have no effect in raising some exception.

However, CODETREK performs worse than CuBERT in EXCEPTION-FUN. This could be attributed to the fact that CuBERT is pre-trained on around 7 million Python programs, and therefore is able to memorize tokens from several instances of try-except blocks. An example of a heuristic that it learns is that in presence of tokens such as request or response in the context, it suggests catching HTTPError, which is usually the correct choice. However, its prediction is not robust against changes in the variable names. For instance, changing the names of a few nearby variables to request or response forces CuBERT to predict HTTPError regardless of the semantics. CODETREK on the other hand, does not rely on memorizing the tokens, but learns to assign high probabilities to walks that correctly traverse a chain of function calls starting from the try blocks to locations in programs (or their libraries) where the exception is originally raised.

CODETREK also performs worse than GREAT in VARMISUSE-FUN. This is because every node that corresponds to a variable is selected to be an anchor for this task. The total number of walks (500 in this task) is divided among these variables. However, there can be hundreds of variables in some programs, resulting in few walks generated for each variable in such cases, diminishing the ability of CODETREK to learn sufficient information about each variable.

## 3.2 ROBUSTNESS OF CODETREK

We evaluate the robustness of CODETREK on additional test data that does not follow the distribution of the training data. This data includes two new datasets: one representing real-world bugs for the VARMISUSE-FUN task and the other consisting of programs mutated using a systematic test-generation framework for the DEFUSE-FUN task. The results are reported in Table 2.

**Real-world bugs.** We manually collect 199 real-world instances containing a VARMISUSE-FUN bug and their corrected counterparts (a total of 398 samples) from commits on GitHub and use them as testing data for the VARMISUSE-FUN task. We define a VARMISUSE-FUN bug as the occurrence of a misused variable that is changed to another in-scope variable in the commit. We evaluate the baselines using this real-world set of bugs. CODETREK outperforms baselines in detecting real-world variable misuse bugs (VARMISUSE-REAL) by achieving an accuracy of 57% which is 6% points better than the second best result obtained by GGNN.

**Mutated programs.** There are several approaches to mutating existing datasets, including transforming existing data (Yang et al. (1992)), generating synthetic programs, and fuzzing. A representa-

tive approach that has been used to systematically evaluate the robustness of compilers is Skeletal Program Enumeration (SPE), proposed by Zhang et al. (2017). SPE parameterizes each program by a set of its variables, and replaces each variable name exhaustively with other in-scope variable names. We generate variations of the DEFUSE-FUN testing data using this technique, and evaluate the baselines on this mutated dataset (DEFUSE-SPE). CODETREK outperforms all baselines in classifying these perturbed programs by achieving the ROC-AUC score of 78% which is 15% points better than the second best result obtained by Code2Seq.

The poor performance of the baselines can be explained by the fact that the code generated by SPE is out-of-distribution. For example, the assignment a = a + a is unusual in real code, but occurs frequently in SPE-generated samples. Despite this, the inductive bias borne by rich relational information during training remains applicable and prevails over the unusual-looking token sequences, thus explaining CODETREK's performance.

These results suggest that sampling walks can be a promising strategy for robustness. Interestingly, the runner-up in this study is Code2Seq—another walk-based approach. We inspected the paths in both approaches to understand the reason behind the difference in performance of Code2Seq and CODETREK despite their similarities. We identified two reasons: 1) the kinds of program information that can be captured from an AST are limited compared to the program graph we propose, and 2) several walks that CODETREK prioritizes for this task cannot be embedded by Code2Seq.

## 3.3 EFFECTIVENESS ON LONGER-RANGE TASKS

We evaluate the effectiveness of CODETREK on tasks that require reasoning beyond function boundaries. CODETREK achieves this ability by readily incorporating relations that capture inter-procedural or inter-modular dependencies such as call graphs. To demonstrate this, we compare CODETREK's performance on EXCEPTION with vs. without incorporating the call graph information at training time. CODETREK achieves an accuracy of 52% when call graph information is not provided, which increases to 63% after providing the call graph information between functions within a module.

```
1  class ZipFile:
2    def __init__(...):
3      self.__check_compression(...)
4    def __check_compression(...):
5      raise NotImplementedError
6  # ...2000 lines of code...
7  class TestZipFile:
8    def test(path):
9      try:
10       zf = ZipFile(path)
11     except [??]:
12       log.warning()
```

Figure 5: Example EXCEPTION task.

To illustrate the kinds of mistakes that the baselines (and also CODETREK in the absence of call graph information) make, consider the representative example in Figure 5, snipped and simplified from the zipfile package. In this example, the model predicts the exception type that should be caught on line 11. However, to make an informed prediction, the model must consider the exceptions that may be raised when calling the ZipFile constructor (line 10). Hence, the definition of the constructor (line 2) must be taken into consideration. This constructor calls another function __check_compression in which a NotImplementedError is raised on line 5. This chain of function dependence can be easily represented using a call graph. Therefore, CODETREK, once provided with call graph, will eventually traverse the path that reaches this raise statement from the exception statement through the call graph edges.

## 3.4 SENSITIVITY TO NUMBER OF SAMPLED WALKS

We evaluate the sensitivity of CODETREK at test time to the number of walks that are sampled from program graphs. All the models for the considered tasks are trained on 100 sampled walks per program. The results are reported in Figure 6. We observe that the accuracies of the models increase with the number of sampled walks. In some tasks, such as DEFUSE and VARMISUSE that involve local reasoning about one point in the program, reducing the number of walks from 100 to 50 reduces the accuracies of the models by a very small amount. On the other hand, for tasks that require reasoning about numerous points in the program (e.g., DEFUSE-FUN) or reasoning globally (e.g., EXCEPTION) decreasing the number of sampled walks has a bigger impact on the accuracy.

## 3.5 IMPACT OF PROGRAM REPRESENTATION

To evaluate the impact of different code representations, we train two models for each task using CODETREK's architecture: for one set of models, the walks are sampled from relational graphs whereas for the other set of models, the walks are sampled from ASTs. The performance results are reported in Table 3. Notably, the models trained on walks sampled from relational graphs are about 3–35% points more accurate than models trained on walks sampled from ASTs.

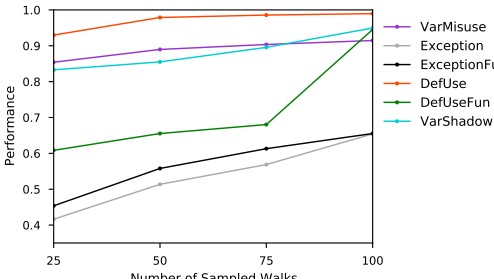

Figure 6: Sensitivity to the number of walks.

| Task | Relational | AST |
|---|---|---|
| VARMISUSE | 0.91 | 0.63 |
| VARMISUSE-FUN | 0.70 | 0.55 |
| EXCEPTION | 0.63 | 0.37 |
| EXCEPTION-FUN | 0.65 | 0.62 |
| DEFUSE ∗ | 0.98 | 0.63 |
| DEFUSE-FUN ∗ | 0.91 | 0.67 |
| VARSHADOW | 0.94 | 0.73 |

Table 3: Impact of Program Representation on Accuracy. (Rows marked by ∗ are evaluated by ROC-AUC score.)

We also evaluate the usefulness of the ability to bias random walks in CODETREK. The accuracy results that are reported in Table 1 are all trained on biased random walks. Specifically, in all of the tasks, nodes with types stmt, expr, and variable are biased such that they are 5 times more likely to be traversed compared to other kinds of neighboring nodes. In addition, in the EXCEPTION task, we decrease the bias assigned to nodes of type module to 0 to avoid traveling from one function to another through the module node they have in common. (See Appendix E for more details about specification of the tasks.) This forces the walks to only go to other functions by taking call graph edges between them. We select one of the tasks, EXCEPTION-FUN, to measure the accuracy in the absence of said biases. We re-train EXCEPTION-FUN using uniformly sampled walks and observe that the accuracy reduces from 65% to 58% as a result.

## 4 RELATED WORK

**Learning to represent code.** There is a rich literature on using neural networks for code reasoning. At the token sequence level, the Transformer and its variants (Hellendoorn et al., 2020; Dowdell & Zhang, 2020) have been widely used (Berabi et al., 2021; Ahmad et al., 2020; Zügner et al., 2021; Kim et al., 2021; Wang et al., 2020). Their performance can be further boosted via pretraining (Feng et al., 2020; Guo et al., 2020; Kanade et al., 2020; Wang et al., 2021; Peng et al., 2021; Liu et al., 2020). Others have proposed to represent programs with ASTs and additional semantic edges (Allamanis et al., 2018; Brockschmidt et al., 2018) or learned abstract relations (Johnson et al., 2020), using GNN or leaf-to-leaf sequence embeddings (Alon et al., 2019a;b). Our work enables adding much richer semantic information while reducing the dependency on syntax structures. Additionally, CODETREK can take advantage of program analysis queries on relational databases to eliminate the engineering burden of augmenting program graphs with additional semantic edges.

**Graph representation learning.** Our work on learning program representations via relational databases is closely related to inductive representation learning on graphs (Hamilton et al., 2017) with graph neural networks (GNNs) (Xu et al., 2018) or Transformers (Ying et al., 2021). Although scalable GNNs via sampling (Chen et al., 2017; Zhou et al., 2020) have been proposed in the transductive setting, it is still challenging to represent large database graphs with 100k nodes in this inductive setting (Clement et al., 2021; Yang & Kuang, 2021). Techniques from transductive graph embedding based on skip-gram (Perozzi et al., 2014; Grover & Leskovec, 2016) or general knowledge graph embedding (Das et al., 2017; Hamilton et al., 2018; Zheng et al., 2020) are scalable but not directly applicable for inductive setting. CODETREK achieves a good balance between modeling for large codebases and efficiency.

**Feature selection techniques.** There are efforts in the data-mining literature to minimize human effort in feature augmentation and selection. Chepurko et al. (2020) discover joins that can improve the prediction accuracy for a single data table whereas CODETREK operates on multiple tables.

## 5 CONCLUSIONS

We proposed CODETREK, a technique that represents programs as relational databases to make rich semantic information available to deep learning models for code-reasoning tasks. We also introduced a flexible walk-based mechanism to sample relevant contexts from large graphs which are constructed from relational databases. We evaluated CODETREK on a variety of real-world tasks and datasets, and showed that it outperforms state-of-the-art neural models.

## ACKNOWLEDGMENTS

We thank the anonymous reviewers, David Bieber, Rishabh Singh, Charles Sutton, and Daniel Tarlow for their valuable feedback. This research was supported by grants from ONR (#N00014-18-1-2021) and NSF (#2107429 and #1836936).

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

# A  TRAINING PARAMETERS AND HYPERPARAMETERS

In this section, we describe details on parameters and hyperparameters we used.

**CodeTrek**   We train CODETREK models with a learning rate of $10^{-4}$, 4 transformer layers, an embedding size of 256, 8 attention heads, and 512 hidden units. We sample 100 walks with lengths of up to 24 in each graph for every task, except for the VARMISUSE-FUN task for which we sample 500 such walks per graph. The reason is that the anchors we select for VARMISUSE-FUN task are all the variables in the given program which can be well over 100 variables. So, we increase the total number of walks to include more random walks starting from each variable.

**CuBERT**   We fine-tune the CuBERT pre-trained model that is provided by Kanade et al. (2020) with a learning rate of $10^{-4}$, 4 transformer layers, and 512 hidden units. We use the checkpoint that is pre-trained on examples of size 512 tokens.

**GREAT**   We train GREAT models with a learning rate of $10^{-4}$, 4 transformer layers, 8 attention heads and 512 hidden units. The example size in 512 tokens.

**Code2Seq**   We train Code2Seq models with a learning rate of $10^{-3}$, 4 layers, 512 hidden units, and embedding size of 256. We sample 100 paths in each AST.

**GGNN**   We train GGNN models with a learning rate of $10^{-4}$, 5 layers, a latent dimension of size 128, and a message dimension of size 128.

# B  MORE EXPERIMENNTS

**Increasing the Number of Layers in Baselines**   We repeat the experiments that are reported in Table 1 for GREAT and GNN with 10 layers. We report the results in Table 4

| Task | GNN | GREAT |
|---|---|---|
| **VARMISUSE** | 0.72 | 0.84 |
| **VARMISUSE-FUN** | 0.58 | 0.86 |
| **EXCEPTION** | 0.30 | 0.45 |
| **EXCEPTION-FUN** | 0.53 | 0.68 |
| **DEFUSE\*** | 0.78 | 0.87 |
| **DEFUSE-FUN\*** | 0.77 | 0.84 |
| **VARSHADOW** | 0.74 | 0.94 |

Table 4: The performance of GNN and GREAT with 10 layers.

**Ablation Study**   We measure the contribution of the positional encoding which is used in embedding walks, the contribution of derived relations in improving the accuracy, and the effect of the biases assigned to node types. We report the results in Table 5. Every row in this table shows a different configuration indicated by 1-4.

We train a model for the EXCEPTION task using the Positional Encoding in embedding the components of each walk, the call relation that shows the relationships between functions and their callers, and the biases which are assigned to nodes of type stmt, expr", and variable (Config 1). This setting is similar to that of Table 1 in the paper. With this setting, CODETREK achieves an accuracy of 63.83% on the EXCEPTION task. If we remove the Positional Encoding (Config 2), we see a small drop of 1.77% points in the accuracy. The effect of further removing the biases (Config 3) is much higher: CODETREK 's accuracy drops 6.3% points from 62.06% to 55.76% points. This aligns with our intuition that adding biases to the aforementioned node types results in generating walks that are more relevant to the task. Finally, we obtain the largest drop in the accuracy by further removing the derived call relation (Config 4). This component contributes a significant amount of 10.57% points to the accuracy of the task, and it obtains a low accuracy of 45.19% points in absence of all three components.

| Config # | Positional Encoding | Derived Relations | Biases | Accuracy (%) |
|:---:|:---:|:---:|:---:|:---:|
| 1 | ✓ | ✓ | ✓ | 63.83 |
| 2 | × | ✓ | ✓ | 62.06 |
| 3 | × | ✓ | × | 55.76 |
| 4 | × | × | × | 45.19 |

Table 5: Contribution of different factors to the accuracy of EXCEPTION task.

| # Steps | 4 | 6 | 12 | 18 | 24 | 30 |
|:---:|:---:|:---:|:---:|:---:|:---:|:---:|
| Accuracy | 0.41 | 0.49 | 0.60 | 0.63 | 0.64 | 0.65 |

Table 6: Sensitivity to the length of walks.

**Different Pooling Mechanisms.** We examine the effect of mean pooling versus attention pooling on the performance of CODETREK models. Attention pooling increases the accuracy of the EXCEPTION task from 63.83% to 66.43%.

**Different Positional Encoding Techniques.** We also explore different positional encoding options. The current setting of CODETREK gives an accuracy of 63.83% in the EXCEPTION task. Substituting the sinusoidal positional encoding with a learned one improves the accuracy less than 1% points. This result is in line with findings of Vaswani et al. (2017), which report nearly identical results using both positional encoding techniques.

**Sensitivity to the Length of Walks** To measure the sensitivity of CODETREK to the length (i.e., number of steps) of walks, we train a number of models for the EXCEPTION task with walks of length 4–30 steps. We report accuracy changes in Table 6. Longer walks tend to improve accuracy. Walks that are too short (4 or 6 hops) result in models with low accuracy (42% and 51%, respectively) because they are not able to capture enough information to make predictions. There is, however, a point when enough context is captured (e.g., 24 hop walks) and longer walks do not improve performance significantly.

## C  CODEQL QUERIES USED FOR LABELING

In this section, we present the CodeQL queries that we used to label the examples for newly added tasks. Both queries are adapted from CodeQL's query repository at `https://github.com/github/codeql`.

### C.1  DEFUSE-FUN QUERY

```
1  import python
2  import Definition
3
4  predicate unused_local(Name unused, LocalVariable v) {
5    forex(Definition def | def.getNode() = unused |
6      def.getVariable() = v
7      and def.isUnused()
8      and not exists(def.getARedef())
9      and not exists(annotation_without_assignment(v))
10     and def.isRelevant()
11     and not v = any(Nonlocal n).getAVariable()
12     and not exists(def.getNode().getParentNode().
13                 (FunctionDef).getDefinedFunction().getADecorator())
14     and not exists(def.getNode().getParentNode().
15                 (ClassDef).getDefinedClass().getADecorator())
16   )
17 }
18
19 private AnnAssign annotation_without_assignment(LocalVariable v) {
20   result.getTarget() = v.getAStore()
21   and not exists(result.getValue())
22 }
23
24 from Name unused, LocalVariable v
```

```
25 where
26   unused_local(unused, v) and
27   forall(Name el | el = unused.getParentNode().(Tuple).getAnElt() | unused_local(el, _))
28 select unused, v.getId()
```

## C.2 VARSHADOW QUERY

```
 1 import python
 2 import semmle.python.types.Builtins
 3
 4 predicate optimizing_parameter(Parameter p) {
 5   exists(string name, Name glob | p.getDefault() = glob
 6                                 | glob.getId() = name
 7     and p.asName().getId() = name
 8   )
 9 }
10
11 predicate shadows(Name d, GlobalVariable g, Function scope, int line) {
12   g.getScope() = scope.getScope()
13   and d.getScope() = scope
14   and exists(LocalVariable l |
15     d.defines(l) and
16     l.getId() = g.getId()
17   )
18   and not exists(Import il, Import ig, Name gd | il.contains(d)
19                  and gd.defines(g)
20                  and ig.contains(gd))
21   and not exists(Assign a | a.getATarget() = d
22   and a.getValue() = g.getAnAccess())
23   and not exists(Builtin::builtin(g.getId()))
24   and d.getLocation().getStartLine() = line
25   and exists(Name defn | defn.defines(g)
26                  | not exists(If i | i.isNameEqMain()
27                  | i.contains(defn)))
28   and not optimizing_parameter(d)
29 }
30
31 AttrNode pytest_fixture_attr() {
32   exists(ModuleValue pytest | result.getObject("fixture").pointsTo(pytest))
33 }
34
35 Value pytest_fixture() {
36   exists(CallNode call |
37     call.getFunction() = pytest_fixture_attr()
38     or call.getFunction().(CallNode).getFunction() = pytest_fixture_attr()
39   | call.pointsTo(result)
40   )
41 }
42
43 predicate assigned_pytest_fixture(GlobalVariable v) {
44   exists(NameNode def |
45     def.defines(v) and def.(DefinitionNode).getValue().pointsTo(pytest_fixture())
46   )
47 }
48
49 predicate first_shadowing_definition(Name d, GlobalVariable g) {
50   exists(int first, Scope scope |
51     shadows(d, g, scope, first)
52     and first = min(int line | shadows(_, g, scope, line))
53   )
54 }
55
56 from Name d, GlobalVariable g, Name def
57 where
58   first_shadowing_definition(d, g)
59   and not exists(Name n | n.deletes(g))
60   and def.defines(g)
61   and not assigned_pytest_fixture(g)
62   and not g.getId() = "_"
63 select d, g.getId(), def
```

## D  CODEBASES AS DATABASES

### D.1  TRANSLATING CODE TO RELATIONAL DATABASE

CODETREK views program information as relations. **??** shows the number of relations in three common programming languages. There is a directed acyclic graph $\mathbb{F}$ that represents the dependencies between these relations. For example, in $\mathbb{F}_{Python}$ represents the dependencies between 95 base relations plus 277 derived relations.

**Example D.1.** A certain Python task requires base relations $\{A, B\}$ and derived relation $\{E\}$. However, in $\mathbb{F}_{Python}$, $E$ depends on derived relation $D$, which in turn depends on base relations $\{B, C\}$. Therefore, Code2Rel computes all five relations: $A$, $B$, $C$, $D$, $E$ in order.

---

**Algorithm 1 (Code2Rel)** Given a program $P$, a set of base relation names $R_B$, and a set of derived relation names $R_Q$, construct and return database $D$. Note that the term *node* in this algorithm refers to nodes in the relation dependency graph, not to *nodes* in the program graph (e.g., in Algorithm 2) that the model will see.

1. Initialize $D$ to the set of all base relations in $R_B$ by translating program $P$.

2. Let $R_S$ be the set of relation names reachable from $R_Q$ in relation dependency graph $\mathbb{F}$:

    (a) $R_Q \subseteq R_S$
    (b) if $r \in R_S$ and $r \to r' \in \mathsf{Edges}(\mathbb{F})$ then $r' \in R_S$

3. Let $F$ be the sub-graph of $\mathbb{F}$ induced by set of nodes $R_S$:

    (a) $\mathsf{Nodes}(F) = R_S$
    (b) $\mathsf{Edges}(F) = \{\, r \to r' \in \mathsf{Edges}(\mathbb{F}) \mid r, r' \in \mathsf{Nodes}(F)\,\}$

4. Compute a topological ordering $L = [r_1, ..., r_{|Nodes(F)|}]$ of $F$.

5. For each $r$ in $L$ in order:

    Evaluate the query for computing relation $r$ on database $D$ and add the result to $D$.

---

### D.2  TRANSLATING RELATIONS TO GRAPH

---

**Algorithm 2 (Rel2Graph)** Given a database $D$, construct a program graph $G$.

Construct an undirected and labeled graph $G$ as follows:

(a) $\mathsf{Nodes}(G) = D$
(b) Add to $\mathsf{Edges}(G)$ each $(t_1, t_2, l)$ that satisfies the following conditions:

  i. $l : R.[a_1, ...a_k] \to S.[b_1, ..., b_k]$ is a referential integrity constraint in the schema of database $D$
  ii. $t_1$ is a tuple of relation named $R$ in $D$
  iii. $t_2$ is a tuple of relation named $S$ in $D$
  iv. for all $i \in [1..k] : t_1.a_i = t_2.b_i$

---

### D.3  TRANSLATING GRAPH TO A SET OF WALKS

**Definition D.1** (Walk specification). A walk specification $S = \langle \mathcal{C}, B, min, max \rangle$ is a tuple in which $\mathcal{C}$ is a conditional expression that filters walk anchors from the set of nodes $\mathsf{Nodes}(G)$, $B$ is a map of bias values that correspond to each relation name, and $min, max \in \mathbb{R}^+$ specify the minimum and maximum length of the walks generated by the specification.

**Example D.2.** An example of a walk specification is as follows.

$\mathcal{C} = \{t | t \in \mathsf{Nodes}(G) \wedge t \in \mathsf{expr} \wedge t.kind = \mathsf{name}\}$
$B = \{\mathsf{stmt} : 5, \mathsf{expr} : 5\}$
$min = 3, max = 16$

**Algorithm 3 (Graph2Walks)** Given a program graph $G$, a walk specification $S = \langle \mathcal{C}, B, min, max \rangle$, and the number of walks $w$, sample a set of walks $W$.

1. Initialize the set of walks $W = \emptyset$.
2. Compute the set of anchors $A = \{t | t \in \mathsf{Nodes}(G) \wedge t \text{ conforms to } S.\mathcal{C}\}$.
3. While $|W| \leq w$:
   (a) Pick a random tuple $t_{curr}$ from $A$.
   (b) Construct $walk$ by repeating the following steps between $S.min$ and $S.max$ times:
       i. Set $t_{prev} := t_{curr}$.
       ii. Set $t_{curr}$ to a $t \in \mathsf{Neighbors}(G, t_{prev})$ with prob. proportionate to $S.B[type(t)]$.
       iii. Let $e_{curr} = (t_{prev}, t_{curr}, l) \in \mathsf{Edges}(G)$
       iv. If $e_{curr} \notin walk$ then extend $walk$ by $e_{curr}$. Otherwise set $t_{curr} := t_{prev}$.
   (c) If $walk \notin W$ then add it to $W$.

## D.4 BRINGING ALL THE PIECES TOGETHER

**Definition D.2** (Task specification). A task specification $T = \langle R_B, R_Q, S, n \rangle$ is a tuple in which $R_B$ is a set of base relation names, $R_Q$ is a set of derived relation names, $S$ is a walk specification as described in Definition D.1, and $n$ is the number of walks to be generated.

**Algorithm 4 (Code2Walks)** Given a program $P$ and a task specification $T = \langle R_B, R_Q, S, n \rangle$, generate a set of walks $W$.

1. $D = \mathsf{Code2Rel}(P, T.R_B, T.R_Q)$
2. $G = \mathsf{Rel2Graph}(D)$
3. $W = \mathsf{Graph2Walks}(G, T.S, T.n)$

## E TASK SPECIFICATIONS

Using the notations defined in Appendix D, we describe the specifications of each task that we used for evaluating CODETREK. In the following specifications, ellipsis (...) indicate the rest of the universe of base relations as designed in Semmle framework.

$T_{\text{VARMISUSE-FUN}} = \{$

$\qquad R_B = \{\mathsf{stmt}, \mathsf{expr}, \mathsf{variable}, \mathsf{ssa\text{-}defn}, \mathsf{ssa\text{-}use}, \mathsf{successor}, ...\},$
$\qquad R_Q = \{\},$
$\qquad S = \{$
$\qquad\qquad \mathcal{C} = \{t \mid t \in \mathsf{Nodes}(G) \wedge t \in \mathsf{expr} \wedge\ t.Kind = \mathsf{name}\},$
$\qquad\qquad B = \{\mathsf{stmt} : 5, \mathsf{expr} : 5, \mathsf{variable} : 5\},$
$\qquad\qquad min = 4, max = 16$
$\qquad \},$
$\qquad n = 500$

$\}$

$T_{\text{EXCEPTION-FUN}} = \{$

$\qquad R_B = \{\mathsf{stmt}, \mathsf{expr}, \mathsf{variable}, \mathsf{ssa\text{-}defn}, \mathsf{ssa\text{-}use}, \mathsf{successor}, ...\},$
$\qquad R_Q = \{\},$
$\qquad S = \{$
$\qquad\qquad \mathcal{C} = \{t \mid t \in \mathsf{Nodes}(G) \wedge t \in \mathsf{stmt} \wedge\ t.Kind = \mathsf{except} \wedge t.Type = \mathsf{HoleException}\},$
$\qquad\qquad B = \{\mathsf{stmt} : 5, \mathsf{expr} : 5, \mathsf{variable} : 5\},$
$\qquad\qquad min = 4, max = 16$

$$},$$
$$n = 100$$

$$}$$

$$T_{\text{EXCEPTION}} = \{$$

$R_B = \{\text{stmt, expr, variable, ssa-defn, ssa-use, successor, ...}\},$
$R_Q = \{\text{call-graph}\},$
$S = \{$
  $\mathcal{C} = \{t \mid t \in \text{Nodes}(G) \wedge t \in \text{stmt} \wedge\ t.Kind = \text{except} \wedge t.Type = \text{HoleException}\},$
  $B = \{\text{stmt} : 5, \text{expr} : 5, \text{module} : 0\},$
  $min = 10, max = 24$
$\},$
$n = 100$

$$}$$

$$T_{\text{DEFUSE-FUN}} = \{$$

$R_B = \{\text{stmt, expr, variable, ssa-defn, ssa-use, successor, ...}\},$
$R_Q = \{\text{variable-defs, local-variables}\},$
$S = \{$
  $\mathcal{C} = \{t \mid t \in \text{Nodes}(G) \wedge t \in \text{expr} \wedge\ t.Kind = \text{name} \wedge t.Context \in \{\text{write, param}\}\},$
  $B = \{\text{stmt} : 5, \text{expr} : 5, \text{variable} : 5\},$
  $min = 4, max = 16$
$\},$
$n = 100$

$$}$$

$T_{\text{DEFUSE}}$ and $T_{\text{VARMISUSE}}$ are defined similar to $T_{\text{DEFUSE-FUN}}$ and $T_{\text{VARMISUSE-FUN}}$, respectively. However, in both $T_{\text{DEFUSE}}$ and $T_{\text{VARMISUSE}}$ their $\mathcal{C}$ has an additional restriction $t.location = $ user selection. Also, $T_{\text{VARMISUSE}}$ is evaluated with $n = 100$.

$$T_{\text{VARSHADOW}} = \{$$

$R_B = \{\text{stmt, expr, variable, ssa-defn, ssa-use, successor, ...}\},$
$R_Q = \{\text{call-graph}\},$
$S = \{$
  $\mathcal{C} = \{t \mid t \in \text{Nodes}(G) \wedge t \in \text{var} \wedge\ t.Kind = \text{name} \wedge t.is\_global = \text{True}\},$
  $B = \{\text{stmt} : 5, \text{expr} : 5, \text{variable} : 5\},$
  $min = 4, max = 16$
$\},$
$n = 100$

$$}$$

# F   WALK FORMAT

We describe the format of a walk as the model views it before encoding using an example. Each walk consists of three lists for node types, node values, and edges.

```
{
    "anchor": "py_stmts(415098,6,415072,2)"
    "trajectory": {
            "node_types": ["py_stmts", "py_scopes",
                "py_Functions", "py_scopes", "py_stmts",
                "py_exprs", "py_exprs", "py_scopes",
                "py_exprs", "py_variables", "v_8"],
```

```
        "node_values": [["ExceptStmt"], "",
            ["pipe","line"], "", ["Assign"],
            ["Call"], ["Attribute"],
            ["Name"], ["metr","ic"]],
        "edges": ["(py_scopes.node,py_stmts.id)",
            "(py_Functions.id,py_scopes.scope)",
            "(py_Functions.id,py_scopes.scope)",
            "(py_scopes.node,py_stmts.id)",
            "(py_exprs.parent,py_stmts.id)",
            "(py_exprs.id,py_exprs.parent)",
            "(py_exprs.id,py_scopes.node)",
            "(py_exprs.id,py_scopes.node)",
            "(py_exprs.id,py_variables.parent)",
            "(py_variables.id,variable.id)"]
    },
}
```

The node_values can consist of the value of any attribute of a node. For instance, ExceptStmt above is the kind of the node with type py_stmts, and pipeline is the name of the function that corresponds to the function which is illustrated by node with type py_Functions. The values that are identifiers, function names, etc. are subtokenized using a subword tokenizer (the tensor2tensor package). For instance, pipeline will break into [pipe,line]. Therefore, each value is encoded using the corresponding vector in the subword dictionary. If the relation name of a tuple is variable, we assign an id to it to be able to distinguish between different variables. So, the type of a variable node is determined by an id (e.g., v_8).

## G  ADDITIONAL INFORMATION ABOUT GRAPHS

### G.1  GRAPH SIZES

| Relational | EXCEPTION-FUN | EXCEPTION | VARSHADOW | VARMISUSE-FUN | DEFUSE-FUN |
|---|---|---|---|---|---|
| average | 5,278 | 802,231 | 62,863 | 1,482 | 1,829 |
| std | 31,550 | 504,758 | 93,248 | 14,623 | 27,503 |
| min | 55 | 585 | 53 | 130 | 139 |
| max | 492,970 | 4,422,586 | 1,285,178 | 483,499 | 612,343 |

Table 7: Number of tuples (i.e., nodes) across relations of each file in the dataset used for each task.

| AST | EXCEPTION-FUN | EXCEPTION | VARSHADOW | VARMISUSE-FUN | DEFUSE-FUN |
|---|---|---|---|---|---|
| average | 176 | 3,202 | 667 | 93 | 177 |
| std | 244 | 4,117 | 1432 | 126 | 268 |
| min | 20 | 20 | 10 | 10 | 10 |
| max | 13,035 | 36,808 | 36,786 | 30,729 | 5,243 |

Table 8: Number of AST nodes for each file in the dataset used for each task.

### G.2  EDGES IN BASELINE GRAPHS

**GGNN Graphs.**  The edges that are represented to GGNN models are borrowed from Allamanis et al. (2018). They include

1. AST edges
2. NextToken edges
3. LastRead/LastWrite/ComputedFrom/LastLexicalUse edges among variable accesses

4. GuardedBy/GuardedByNegation edges between variables used in branches and their corresponding conditional expressions

5. ReturnsTo edges from the return statement to the method declaration, and

6. FormalArgName edges between method call arguments and their corresponding formal parameters.

**GREAT Graphs.** For GREAT models, similar to Hellendoorn et al. (2020), we borrow the edge types from Allamanis et al. (2018) and augment them with function calls.

## H DATASET SIZES

The number of samples in each dataset is shown in Table 9. The VARSHADOW dataset consists of 49% samples with positive labels. The DEFUSE-FUN dataset is more skewed, with 15% positive labels, while the DEFUSE dataset has 1.6% positive labels.

| Task | # Training | # Validation | # Testing | Avg. LoC ($<$max) |
|---|---|---|---|---|
| VARMISUSE | 700,683 | 75,468 | 378,401 | 13 ($<$235) |
| VARMISUSE-FUN | 700,683 | 75,468 | 378,401 | 13 ($<$235) |
| EXCEPTION | 18,456 | 2,086 | 10,334 | 528 ($<$7,624) |
| EXCEPTION-FUN | 18,456 | 2,086 | 10,334 | 32 ($<$1,835) |
| DEFUSE | 217,591 | 52,598 | 104,111 | 12 ($<$528) |
| DEFUSE-FUN | 33,182 | 8,149 | 16,296 | 12 ($<$528) |
| VARSHADOW | 70,183 | 21,794 | 39,845 | 149 ($<$27,228) |

Table 9: The number of samples used for training, validation, and testing and the lines of code that they contain. Lines of code (LoC) is reported as the average lines of code across samples in each dataset after removing the highest 0.1% and lowest 0.1% of the data.

## I QUALITATIVE STUDY

We qualitatively discuss a few examples on specific code snippets and describe the walks that contribute the most to the predictions made by the respective CODETREK models. In each example, we explain how the relations and the semantic edges between them enable CODETREK's models to predict accurately.

### I.1 EXAMPLE 1 (DEFUSE)

```
1  def write_random_to_file():
2      no = random.randint(1, 10)
3      with open("random.txt", "w") as file:
4          file.write(str(no))
5      return no
6
7  def write_random():
8      random_no = write_random_to_file()
9      print "A random number was written to random.txt"
```

Figure 7: A sample code snippet for DEFUSE.

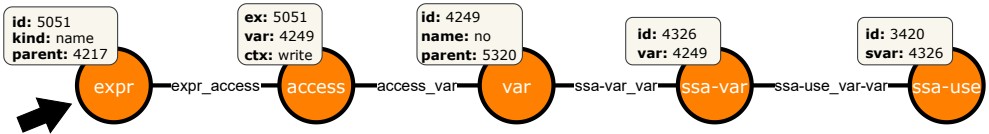

Figure 8: The most important walk in a simple instance of DEFUSE.

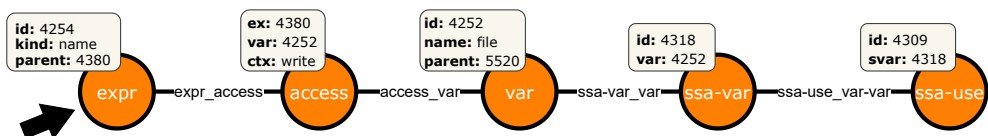

Figure 9: The most important walk in a challenging instance of DEFUSE.

To understand how CODETREK uses semantic relations to determine whether a defined variable is used, consider the code snippet listed in Figure 7. In this snippet, the local variable file is defined on line 3, and then used on line 4. Intuitively, one would start from the variable definition and follow the code to find an access of it to prove that it is indeed a used variable. More specifically, a programmer starts with the expression on line 3 in which variable file is defined. She then tries to find another access to this variable that reads it, such as on line 4.

CODETREK determines that the walk illustrated in Figure 8 has the highest score among a set of randomly generated walks. Interestingly, this walk shows a similar behavior to that of a programmer: it starts at the anchor node (an expr node corresponding to the variable definition) which corresponds to the expression that defines file. Then, it traverses the graph towards a node that corresponds to a use of this variable (a ssa-use node corresponding to the variable use).

```
1  def construct_file_handle():
2      file = Handler.initialize()
3      def check_handle(file):
4          if file.id < MIN_H:
5              return False
6          return True
7      file = Handler.initialize()
8      return Handler.default()
```

Figure 10: A challenging code snippet for DEFUSE.

Even the models that do not embed semantic edges (e.g., CuBERT) are able to correctly predict that file is used, in such simple cases. However, in more complicated cases, such as the code snippet listed in Figure 10, semantic edges are needed to be able to distinguish between different definitions of variable file and to not confuse various uses of them. In this snippet, file is defined on line 2, and then on line 7. The file defined on lines 2 and 7 are never used. To make matters more complicated, there is a function check_handle that is defined inside the top-level function construct_file_handle. This function takes an argument which is named file and uses it on line 4.

In the absence of edges that make the relationship between uses and definitions of variable explicit, it is challenging for a model to determine that the variable named file on line 2 is different from the variable of the same name on line 4. As a result, we see that GREAT and CuBERT fail to label the variable definition on line 2 as unused. CODETREK, however, takes advantage of the relationship between the variable definition and its use (an ssa-use node) and makes a robust prediction. Among the set of randomly generated walks starting from the definition on line 2 (expr node with id 4244) there are no walks with the following pattern which only occurs when a variable is used after being defined: "expr → access → var → ssa-var → ssa-use". Therefore, CODETREK predicts that this variable is never used. On the other hand, as illustrated in Figure 9, a walk with the mentioned pattern exists between the definition on line 3 (expr node with id 4254) to its use on line 4 (ssa-use node with id 4309). So, CODETREK predicts that this variable is used. It is worth emphasizing that the walks illustrated in Figure 8 and Figure 9 are very similar although they correspond to completely different code snippets.

## I.2  EXAMPLE 2 (EXCEPTION)

For the EXCEPTION task, we choose a code snippet from the test dataset, which is listed in Figure 11. In this code snippet, CODETREK must predict the exception to be caught by the except statement at line 36 (represented by HoleException). The correct exception is ValidationError. We know this

```
1   class Admission:
2
3       # ...
4
5       def admit_car(self, car):
6           if not str(car.get_id()).isdecimal():
7               raise ValidationError("Index is not valid")
8           name = car.get_number()
9           if name.upper() != name:
10              raise ValidationError("Number not in capslock")
11          if name.count(" ") < 2:
12              raise ValidationError("Number should have 3 parts")
13          check_name = name.split(" ")
14          if not check_name[0].isalpha():
15              raise ValidationError("First part is not alpha")
16          if not check_name[1].isdecimal():
17              raise ValidationError("Second part is not decimal")
18          if not check_name[2].isalpha():
19              raise ValidationError("Third part is not alpha")
20          owner = car.get_owner().replace("-", " ")
21          if not owner.isalpha() or not owner.istitle():
22              raise ValidationError("Owner's name is not written correctly")
23          if len(owner) > 40:
24              raise ValidationError("Name too long")
25
26  # a number of other unit test functions removed here only for presentation purposes ...
27
28  def test_car_admit():
29      admit = Admission.get_instance()
30      car1 = Car(1, "ag 12 BOB", "Dan")
31      car4 = Car(4, "A", "Ian")
32
33      try:
34          admit.admit_car(car1)
35          assert False
36      except HoleException:
37          assert True
```

Figure 11: A sample code snippet for EXCEPTION.

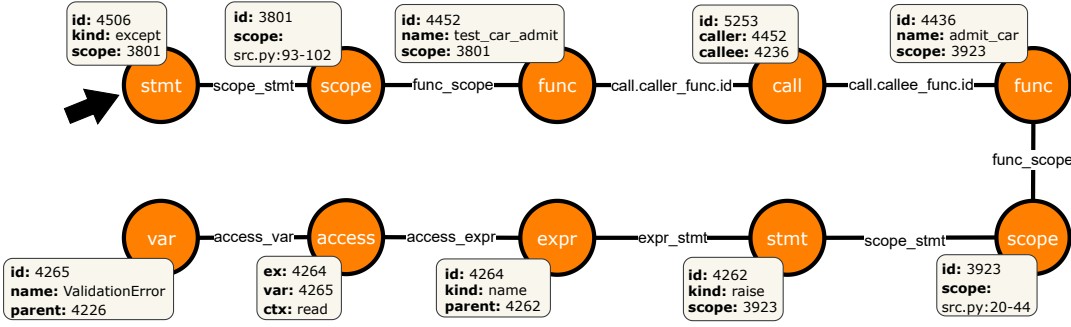

Figure 12: The most important walk in an instance of EXCEPTION.

because we observe that the function admit_car is called in the corresponding try block, and upon inspecting its definition, we see that it raises the ValidationError exception.

Out of the walks sampled by CODETREK for predicting the correct exception, we illustrate the most important (highest scoring) walk in Figure 12. This walk represents the aforementioned intuitive reasoning for predicting the exception. It starts at the anchor node, which is the node of type stmt with id 4506, corresponding to the except statement on line 36. It traverses to the function definition of admit_car by first traversing to the call node for admit_car with id 5253, representing the call on line 34, and then following the corresponding call graph edge to the definition of admit_car. These call graph edges allow for such inter-procedural reasoning. The walk then traverses to the stmt node for the raise statement, then to its expression, and reaches the ValidationError exception via its corresponding access node.

## I.3 EXAMPLE 3 (VARSHADOW)

```
1  env_vars = env.vars
2
3  class SystemReq:
4    # ...
5
6  # ...
7
8  class Utils:
9    @staticmethod
10   def rev(s):
11     for i in range(len(s)//2):
12       tmp = s[i]
13       s[i] = s[-(i+1)]
14       s[-(i+1)] = tmp
15
16   @staticmethod
17   def env_check():
18     env_vars = environ.vars
19     return env_vars
```

Figure 13: A sample code snippet for VARSHADOW.

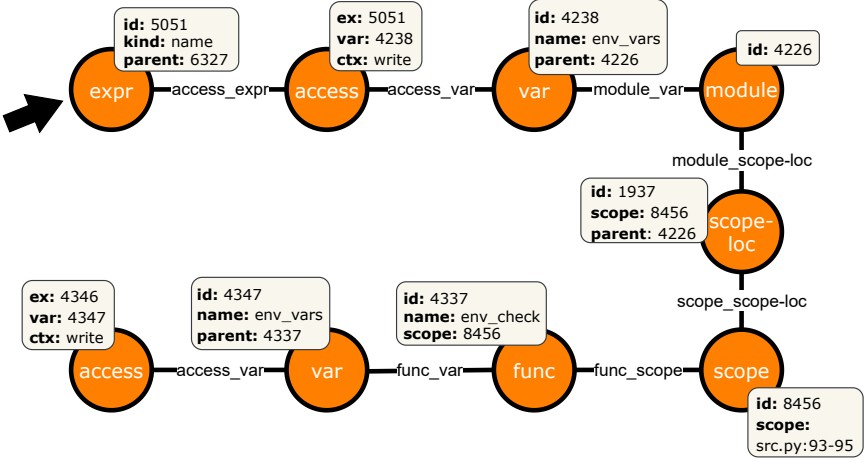

Figure 14: The most important walk in an instance of VARSHADOW.

VARSHADOW is an example of a long-range task in which the model has to be able to distinguish between the global and the local scopes in order to predict whether a global variable is shadowed by another variable with the same name that is defined in a local scope. We use the code snippet in Figure 13 to explain how semantic relations help in such tasks.

To determine whether a global variable is shadowed by a local variable, a programmer would look for variables that are defined in local scopes and have the same name as the global variable. The walk which is illustrated in Figure 14 captures the relationship between the global variable definition (expr node with id 5051) and a local re-definition with the same name (access node with id 4346) by visiting a local scope (scope node with id 8456) of the module (the module node) along the way. Interestingly, CODETREK assigns the highest importance to this walk among a number of randomly generated walks, and can therefore correctly predict that env_vars is a shadowed global variable.

