# OpenReview forum: "CodeTrek: Flexible Modeling of Code using an Extensible Relational Representation"
_ICLR.cc/2022/Conference — ICLR 2022 Poster_

### Official Review · Reviewer_okqQ · 2021-10-26

**Correctness:** 3
**Technical Novelty And Significance:** 2
**Empirical Novelty And Significance:** 2
**Recommendation:** 5
**Confidence:** 5

**Main Review:**

## Pros
+ The proposed system shows strong empirical results.
+ The system is compared to a variety of structural and textual baselines.

## Cons
- The paper is very difficult to understand. Many details are not well defined nor explained. See below.

- Ease of use - the system has many "moving parts" that need to be manually designed. While this can be a good thing for experts, it limits the system's applicability to non-expert users and hinders its generalization to other domains, tasks, and languages. The authors claim that "A task developer need not be a machine-learning expert to bring in more semantic information about programs". However, there are many more aspects that the task developer needs to consider: which queries to write, how to write these queries, how to choose the starting node, how to bias the random walk, etc. These might make the proposed system to be very difficult to use for other tasks and languages.

- The paper describes a specific system, and its contributions in terms of machine learning are relatively minor. The paper also lacks a deeper analysis or an ablation study to pinpoint the exact contributing factors and discuss their alternative, nor concrete examples which will help readers to better understand the impact of the proposed method.


### Clarity

* The paper mentioned Semmle several times and mentioned that Semmle converts codebases into relational databases. It is not explained what is Semmle? How does Semmle work? How does it convert codebases into databases? Is it a commercial product?

* Figure 1 is very nice visually but is still very difficult to understand. Some details are explained at the beginning of Section 2, but are still very hard to follow. If I did not miss anything,
other details in this very detailed figure are not explained anywhere. Some explanations that do exist are very hard to follow and to connect with the figure, for example:

>"The derived information is stored in def, which, together with call, can bias the prediction of the best variable to replace a placeholder"

I am not sure how to understand this sentence and what in Figure 1 should I look at.
This unclarity continues in Section 2, where no clear definitions nor explanations are provided. The paragraph "Code as a graph" is very unclear, and frequently uses terms that were not defined.


**Summary Of The Paper:**

The paper presents a framework called CodeTrek for processing and learning source code. The main idea is to represent code as a graph, manually design analyses to serve as additional relations, and perform a set of random walks on the graph. These walks can also be manually guided. Finally, the set of walks is encoded to represent program elements.

**Summary Of The Review:**

Overall, the paper shows strong empirical results. However, I believe that this is an application with little novelty, and few lessons to be learned for the ICLR audience. Hence, I would tend to reject this in favor of more ML-heavy papers.

---

> ### Author Response · Authors · 2021-11-15
> **Author Response to Reviewer okqQ**
>
> **1. Addressing the lack of clarity**
>
>
> We have attempted to clarify several points that the reviewer mentions in the revised version. We have broken the original Figure 1 into Figures 1, 2 and 3 in the revised version to make it easier to understand. Figure 1 now shows, in more detail, how Semmle works to convert codebases into databases. Also, while Semmle is indeed a commercial product, it did not hinder our generation of the databases since their language schemas and queries are publicly available and well-documented.
>
> We have also tried to clarify the Code as a Graph paragraph in the revised version.
>
> **2. Regarding the ease of use**
>
> Given a new code reasoning task, even in the absence of expert knowledge, there are very few changes that are required. There is no need to design and write any new queries; CodeTrek uses a set of fixed, pre-written CodeQL queries. For the rest of the parameters, we introduce a “default specification” in the paper to reduce the engineering effort. This allows us to concretely define new tasks without a lot of effort, such as:
>
> * _“Given a program, predict whether all its call statements have the correct number of arguments.”_ In this task, since the prediction is made about `“call”` statements, the anchors are any call statements. For the length of the walks, the number of the sampled walks, and the bias, the “default specification” that we introduce in the paper can be applied.
> * _“Given a function, predict a suitable name for it.”_ In this task, since the prediction is made about the entirety of a function, we do not need to specify any anchor nodes; instead, multiple randomly selected nodes in the function body can serve as the anchors. Again, in this task, we use the default specification for the walk length, the number of sampled walks, and biases.
>
> Likewise, applying CodeTrek to other languages can be done with little engineering effort since Semmle provides a fixed set of language-specific relations for other languages (e.g. Java, C++, JavaScript, etc.) which can be used instead of Python’s base relations.
>
> **3. "The paper describes a specific system..."**
>
> We acknowledge that our contributions primarily involve the program representation used for reasoning over code and we thank the reviewer for pointing this out. We are currently running ablations to measure the contribution of different factors such as the positional encoding and the length of the walks as well as an alternative pooling mechanism. We will report the results and include a discussion of the ablation studies as soon as they are ready.

---

> > ### Comment · Reviewer_okqQ · 2021-11-30
> > **End of discussion period comment**
> >
> > Dear authors,
> >
> > Thank you for your efforts and responsiveness.
> >
> > The proposed system shows strong empirical results. However, I still think that the paper describes a too specific system, powered by carefully engineered features (in the form of edges and relations), and which is tailored to a specific engine (Semmle).
> >
> > I was left with too many questions, such as:
> >
> > * If the general idea is to use a backend static analysis engine, can this generalize to other engines?
> > * What if we could use only *some* of the relations, e.g., as in Table 2 in Allamanis et al. 2018 (I can imagine that if we do not have access to the entire project, some relations could not be extracted)?
> > * And if we do have all the kinds of relations, which of them are the most important?
> >
> > These questions are important to answer to allow future work to build on the principles of this paper in other tasks, programming languages, and using other static analysis tools.
> >
> > Although it might be a useful system, I think that the paper could have more impact on the software engineering community.
> >
> > In my review, I also commented that the paper lacks concrete examples which will help readers to better understand the impact of the proposed method, and gain some intuition regarding why the semantic relations help. Unfortunately, the authors did not include more examples. More examples, possibly with a qualitative analysis of them, could significantly strengthen the paper.

---

> ### Author Response · Authors · 2021-11-17
> **Author Response to Reviewer okqQ (Part 2)**
>
> > **The paper describes a specific system, and its contributions in terms of machine learning are relatively minor. The paper also lacks a deeper analysis or an ablation study to pinpoint the exact contributing factors and discuss their alternative...**
>
> We performed ablations to study the contribution of different factors for one of our inter-procedural tasks, Exception.
>
> We measured the contribution of the positional encoding which is used in embedding walks, the contribution of “derived relations” in improving the accuracy, and the effect of the biases assigned to node types. We report the results in Table A. Every row in this table shows a different configuration indicated by 1-4.
>
> Config#|Positional Encoding|Derived Relations|Biases|Accuracy|
> -------|:-----------------:|:---------------:|:----:|:------:|
>    1   |       yes         |       yes       | yes  |  63.83 |
>    2   |        no         |       yes       | yes  |  62.06 |
>    3   |        no         |       yes       |  no  |  55.76 |
>    4   |        no         |        no       |  no  |  45.19 |
> _(Table A)_
>
> We train a model for the Exception tasks using the Positional Encoding in embedding the components of each walk, the `“call”` relation that shows the relationships between functions and their callers, and the biases which are assigned to nodes of type `“stmt”`, `“expr”`, and `“variable”` (Config 1). This setting is similar to that of Table 1 in the paper. With this setting, CodeTrek achieves an accuracy of 63.83% points on the Exception task. If we remove the Positional Encoding (Config 2), we see a small drop of 1.77% points in the accuracy. The effect of further removing the biases (Config 3) is much higher: CodeTrek’s accuracy drops 6.3% points from 62.06% to 55.76% points. This aligns with our intuition that adding biases to the aforementioned node types results in generating walks that are more relevant to the task. Finally, we obtain the largest drop in the accuracy by further removing the derived `“call”` relation (Config 4). This component contributes a significant amount of 10.57% to the accuracy of the Exception task, and it obtains a low accuracy of 45.19% points in absence of all three components.
>
> We also performed a sensitivity analysis on the Exception task by varying the length of walks. We report the performance in the Table B below. Increasing the length of the walks generally increases the accuracy. Walks that are too short (length=4 or length=6) result in models with very low accuracy (40.59% and 49.10% points, respectively) because they are not able to capture enough information to make predictions.  On the other hand, there is a point where enough context is captured (e.g., by generating walks with length 24) and longer walks (e.g., walks with length 30) do not contribute significantly to the performance.
>
> | Walk Length | Accuracy |
> |:-----------:|:--------:|
> |           4 |    40.59 |
> |           6 |    49.10 |
> |          12 |    60.05 |
> |          18 |    63.13 |
> |          24 |    63.83 |
> |          30 |    63.96 |
> _(Table B)_
>
> Finally, we compared different pooling mechanisms that can be used in CodeTrek for embedding walks. The reported result for the Exception task (in the paper) is obtained from a model that uses the mean pooling mechanism over the output of the Transformer (Figure 3 in the revised draft). We tried an alternative pooling mechanism, attentional pooling, instead of the mean pooling and obtained an accuracy of 66.43% points which is a 2.6% points improvement over the model that uses the pooling mechanism.

---

> > ### Comment · Reviewer_okqQ · 2021-11-18
> > **Thank you for your answers**
> >
> > Thank you for your answers.
> >
> > I think that more examples for all tasks will significantly benefit the paper, along with the predictions made by each model and baseline.
> > I think that the main feeling that this paper gives the reader is that it is unclear whether this really works, how, and in which cases.
> >
> > I have read the authors' discussion with Reviewer n7bP, who is concerned regarding the validity of the experimental results.
> > I am following that discussion and I will adjust my recommendation accordingly.

---

### Official Review · Reviewer_3Z8h · 2021-10-30

**Correctness:** 2
**Technical Novelty And Significance:** 2
**Empirical Novelty And Significance:** 2
**Recommendation:** 5
**Confidence:** 4

**Main Review:**

This paper is well written and the analysis of the experimental results are sufficient. But I have some questions to discuss.

(1) The novelty in the network architecture is not enough. CodeTrek spends too much weight on the description of Code2Rel and Rel2Graph procedure, however, the network architecture i.e., Transformer is simple. Furthermore, the idea of sampling walks is the same with Code2seq and the difference is mainly on the constructed graph/AST is different.

(2) This paper conducts some sensitivity analysis on the number of the sampled walks, however, the discussion of another important factor walk length (24 in the appendix) is missed especially for the exception task, which may be critical. So I would like to see the performance.

(3) CodeTrek selects the anchor node or in other words, assign higher probabilities to some specific node types for example stmt, expr, it would be great to provide more insights on why choosing these nodes as the start point.

(4) In Rel2Graph, there are two types of relations i.e., base relations and derived relations. Derived relations are customized by the program analysis queries on the base relations, so it can be regarded as another featuring engineering. Is there a way to only use the base relations to construct the graph and see the experimental results? It would be important to see how much the derived relations weigh.

(5) In Table 1, the gap between GGNN and GREAT on Varmisuse is very obvious (0.69 VS 0.82), however, in the original Hellendoorn et al. [1], the gap is not obvious GGNN VS GREAT (0.792 VS 0.769). So what is the reason behind this, since the different dataset were used or the different graph structures?

[1] GLOBAL RELATIONAL MODELS OF SOURCE CODE. Hellendoorn et al.



**Summary Of The Paper:**

This paper proposes a new program representation approach CodeTrek, which leverages a program analysis tool (Semmle) to produce the rich representation of the context for a program. Semmle is able to convert the program into a relational database that can capture the semantics behind the program, furthermore, it also supports extracting the task-specific semantics with the query language CodeQL to represent new semantic information. Then CodeTrek utilizes a biased graph-walk mechanism for pruning the context paths and feeds them to the transformer encoder to learn the path representations and followed by deepset to get a vector representation. The extensive experiments on four diver Python tasks: variable misuse, exception prediction, unused definition and variable shadowing prove that CodeTrek can outperform the current baselines i.e., Code2seq, GGNN, GREAT, CuBERT by a significant margin.

**Summary Of The Review:**

This paper proposes a new code representation technique that is based on the relational database to capture the program semantics for different tasks, however, the novelty in the model architecture is not enough and the proposed code representation approach is more like a feature engineering.

---

> ### Author Response · Authors · 2021-11-15
> **Author Response to Reviewer 3Z8h**
>
> We are currently running the ablation studies to examine the effect of different walk lengths on the performance of the model. We will report the results as soon as they are ready.
>
> > **(1) The novelty in the network architecture ...**
>
> The current neural networks in CodeTrek rely on pre-existing architectures, as pointed out by the reviewer. Exploring novel network architectures is indeed an interesting topic. However, this work aims to take advantage of the wealth of program analysis literature, uniformly and in a systematic manner, in order to use neural networks more effectively in modeling code. The current method is a design choice, but we hope to open up the new opportunities for exploring more neural models.
>
> There is a key difference between the walks that CodeTrek’s walk generator samples and that of Code2Seq: CodeTrek allows the walks to start and end at any arbitrary type of node (e.g., `“stmt”`, `“location”`, etc.). On the other hand, Code2Seq considers paths between AST leaves to encode information about the root of that AST. This difference allows CodeTrek to reason about any arbitrary program element by considering the code surrounding that element, its dependents as well as its dependencies, while Code2Seq’s context is limited to the descendant AST nodes of that particular element. This larger context available to CodeTrek helps it perform better than Code2Seq on all the tasks.
>
>
> > **(2) This paper conducts some sensitivity analysis ...**
>
> We are currently running the ablation studies to examine the effect of different walk lengths on the performance of the model. We will report the sensitivity of the results to this factor as soon as it is ready.
>
>
> > **(3) CodeTrek selects the anchor node ...**
>
>
> _Biases:_ The primary purpose of biases over nodes is to guide the walk generator to sample more relevant walks by reducing the probability of traversing nodes that may diverge from the relevant portions of the graph. For example, if a walk was to traverse to its parent `“module”` node, it may then traverse to other child nodes within the module that may not be relevant to the problem. On the other hand, node types like `“expr”`, `“stmt”`, and `“variable”` tend to be more relevant while also relating to other useful nodes. Hence we give these node types a higher bias in the default specification than other node types. We do examine the effect of the absence of these biases on the Exception-Fun task in Section 3.4.
>
> _Anchors:_ Anchor nodes are selected based on the task definition. For instance, if the task is to predict whether a variable access is correct, then nodes that correspond to “variable accesses” become the anchors. As another example, if the task is to predict the exception type that is thrown, then the nodes that correspond to except statements would become anchors. In cases where the task concerns the whole program, multiple nodes would be chosen randomly as the anchors.
>
> > **(4) In Rel2Graph, there are two types ...**
>
> The graphs for all of our tasks are constructed only from base relations. Only in the exception task, where an inter-procedural analysis is required, we additionally use a derived relation `“call”` to capture the dependencies between functions and their callers. In Section 3.3, we remove this derived relation to measure its contribution. We see that the accuracy drops from 63% to 52%.
>
>
> > **(5) In Table 1, the gap between GGNN and GREAT ...**
>
> The reported gaps are different because the size of the models that [1] train is different from our trained models. Moreover, in [1], the training is done on functions with up to 250 tokens whereas we perform the training on all the functions from Py150ETH Open evaluation dataset (Kanade et al., ICML 2020) without any limitation on the number of the tokens.
>
> We also tested both models on a different set of 398 manually collected real-world VarMisuse bugs and saw an insignificant gap (2%) between GGNN and GREAT. This aligns more with the findings of Hellendoorn et al. (ICLR 2020) on real-world VarMisuse bugs as well as the reviewer’s observations. We have presented the results of the evaluation on this dataset in Section 3.2 and Table 2.

---

> ### Author Response · Authors · 2021-11-17
> **Author Response to Reviewer 3Z8h (Part 2)**
>
> > **(2) This paper conducts some sensitivity analysis on the number of the sampled walks, however, the discussion of another important factor walk length (24 in the appendix) is missed especially for the exception task, which may be critical. So I would like to see the performance.**
>
> We performed a sensitivity analysis on the Exception task by varying the length of walks. We report the performance in the table below. Increasing the length of the walks generally increases the accuracy. Walks that are too short (length=4 or length=6) result in models with very low accuracy (40.59% points and 49.10% points, respectively) because they are not able to capture enough information to make predictions.  On the other hand, there is a point where enough context is captured (e.g., by generating walks with length 24) and longer walks (e.g., walks with length 30) do not contribute significantly to the performance.
>
> | Walk Length | Accuracy |
> |:-----------:|:--------:|
> |           4 |    40.59 |
> |           6 |    49.10 |
> |          12 |    60.05 |
> |          18 |    63.13 |
> |          24 |    63.83 |
> |          30 |    63.96 |

---

> > ### Comment · Reviewer_3Z8h · 2021-11-19
> > **Thanks for the responses**
> >
> > Thanks for the responses.
> >
> > I have also read other reviewers' comments and responses.  In this round of revision, the authors have added the details about the evaluation baseline. For GGNN, the authors claim that they built the graph following Allamanis et al. One more question is that is there an existing tool to help build this graph for python language (if yes, can you share the tool ) or did the authors build this graph from the scratch?

---

> > > ### Author Response · Authors · 2021-11-22
> > > **Author Response to Reviewer 3Z8h**
> > >
> > > To the best of our knowledge, there is no existing tool publicly available to build the graph proposed by Allamanis et al. for Python. So, we had to build these graphs from scratch. In fact, we believe that such engineering efforts are one of the important aspects that differentiates other graph-based representations of code from CodeTrek.

---

### Official Review · Reviewer_yHJY · 2021-11-02

**Correctness:** 4
**Technical Novelty And Significance:** 2
**Empirical Novelty And Significance:** 2
**Recommendation:** 5
**Confidence:** 5

**Main Review:**

Pros:
* A general framework for generating a relational representation of programs for programming-related tasks
* The use of “declarative” tasks (which can be solved without a learned model, e.g. variable shadowing) to evaluate framework/approaches.
* A general methodology that specifies a TODO list (defining queries, random walk bias, anchors, etc.) for tackling a new task (it doesn’t mean that these steps are easy).
* The use of long-distance and cross modules dependencies to solve the tasks.


Cons:
* This paper uses standard neural architectures and the representation that is used is also rather low on novelty.
* A lot of effort is required in preprocessing, before feeding the NN. This includes an expert’s knowledge to determine task-specific queries and task-specific random walk bias (although these were not used in the experiments). To conclude, it feels like that given a new task, it’s not straightforward to apply this approach.
* This paper has less of a contribution to the ML community but has a potential impact on the SE community. The only ML novelty I can spot here is the use of the graph to represent the schema, which can be then processed by some NN.
* This approach is limited to cases where one has the whole project in hand, and the ability to perform (potentially heavy) static analysis of the code.

* Allamanis et al. have shown how to represent programs as graphs, including cases when the program information is enriched by a wide range of semantic information from program analysis. They used the resulting graph representation as input to GNNs, but there’s nothing that precludes the same input from being linearized (by standard graph walks) and used in sequential models.

* Given the expressiveness of the framework, I was hoping for evaluation on tasks that require a more global and long-range kind of properties. Unfortunately, most of the paper focuses on tasks that have a local (intra-procedural) solution. While I understand this from the need to compare to existing baselines, I think that it does not play to your strengths.

* In your evaluation, you only experimented with classification tasks. It will be beneficial to see other kinds of tasks such as summarization and code generation.

* You mention in several places a “walk score”, but didn’t define it, and how it’s calculated and used.

* Table 3 is interesting. Does it mean that the expected benefit from the deep and often complicated semantic analysis is sometimes as low as 20%? Do you think sampling more paths from the AST could have narrowed the gap?


**Summary Of The Paper:**

The paper presents a framework for creating task-specific program representations for deep learning based on program analysis and random graph walks. The basic idea is to use program analysis to create a graph-based representation of the program, capturing both syntactic and semantic relationships between elements. This graph-based representation of the program is then encoded by embedding a set of (random) walks in the graph.
The paper evaluates the framework using four tasks: three tasks that only require intra-procedural program analysis (var-misuse, infer exception type, predict where a definition is used) and one that required inter-procedural analysis (variable shadowing).
CodeTrek is shown to outperform existing techniques in most benchmarks.


**Summary Of The Review:**

This paper proposes a general framework for applying deep learning (specifically DeepSet + Transformers) to code. The authors suggest the following steps for handling code-related tasks:
Building a relational database based on the code by applying pre-defined queries and user/task-specific queries.
Converting the database into a graph, defining some user/task-specific random walk bias and length, and user/task-specific anchor nodes.
Applying this random walk, encoding these using Transformers, and feeding the result to some task-specific neural network.

Overall I think it’s a good paper with a minor ML contribution. It shows that the use of the relational graph outperforms the common use of AST and data flow graphs. It has its drawbacks  - heavily dependent on expert knowledge per language and per task. Further, even experts can’t tell which are the best queries/random walk for the given task.

---

> ### Author Response · Authors · 2021-11-15
> **Author Response to Reviewer yHJY (Part 1)**
>
> **1. Addressing the lack of ML novelty**
>
> While we acknowledge that the ML novelty lies primarily in the program representation used, we still believe that it could be a worthwhile contribution to the ML community. It has been shown in previous work on GGNN and GREAT that the performance of neural models for code can be improved significantly by incorporating features obtained using static analysis.  However, existing systems use a very ad hoc approach while incorporating the results of these analyses. Furthermore, these analyses are often difficult to implement correctly, and so extending existing models with more analyses requires significant engineering. As a result, there is a gap between the static analysis literature and its use in neural models.
>
> CodeQL provides a rich library of high-quality static analyses written by experts whose results can be incorporated in a uniform and systematic representation. Our contribution is, therefore, to construct such a representation that bridges the aforementioned gap in order to reduce the efforts required to incorporate static analysis results in neural models and extend existing models with new analyses.
>
>
> **2. How difficult is it to apply CodeTrek to new tasks?**
>
> We want to clarify that all the queries used by CodeTrek are fixed pre-written queries, and CodeTrek does not require queries to be tailored to each task. As the reviewer points out, the experiments do not use task-specific queries, but instead use the same universal set of base relations. Only the Exception task requires a single additional derived relation `“call”` in order to enable CodeTrek to perform inter-procedural analysis.
>
> We do acknowledge that while CodeTrek shows promising improvements in performance without using task-specific biases, determining the optimal bias for each task is an open problem and an area of research we intend to explore in the future.
>
>
> **3. Addressing "This approach is limited to ..."**
>
> The current approach is not limited to entire projects; it only requires that the code be syntactically correct so that Semmle can statically analyze it. This is the case for all the tasks since they provide only individually defined functions (or modules in the case of Exception and VarShadow tasks) as inputs to Semmle, rather than entire projects. A similar requirement also exists for other state-of-the-art tools like GGNN and GREAT since they too need to statically analyze code to build their representations. The static analysis of the code itself just relies on the ability to install and run the command line interface of CodeQL, since the analyses are all in the form of pre-written CodeQL queries. This setup itself does not require much effort.
>
>
> **4. Difference between Allamanis et al. and CodeTrek**
>
> The reviewer raises an interesting point regarding linearizing the graphs using edge types introduced in Allamanis et al. However, we note that the edges which result from the base relations in CodeTrek are a superset of the edges used in those graphs. Additionally, since the types of those edges are few and hand-engineered according to their specifications, extending these edges with results of additional static analyses in the literature would require significant engineering effort.  Examples of additional static analyses include the function call graph (from the `“call”` derived relation in Figure 1) for the Exception task, information about scopes and basic blocks for the ShadowVariable task, and SSA dependencies for the DefUse and VarMisuse tasks.
>
> We also note that since the analysis performed in Allamanis et al. was for C# code, using those edges for Python code required reimplementing the same static analyses from scratch over Python’s AST libraries, as was done for GREAT by the authors for Hellendoorn et al. However, CodeTrek leverages Semmle’s schema and publicly available queries to detach the work of static analysis from analyzing each language individually with custom libraries.
>
> **5. Regarding long-range tasks**
>
> In our evaluation, we have used two long-range tasks and two local tasks. Our long-range tasks are the Exception task and global variable shadowing (VarShadow) task. In both of these tasks, the distance between the program elements that need to be analyzed/considered can be hundreds of lines of code because they can be located in different classes that are located far from each other (e.g., hundreds of lines of code away). In addition, the VarShadow task indeed requires a global view of the source code to determine whether a global variable is used in any of the multiple inner scopes.

---

> ### Author Response · Authors · 2021-11-15
> **Author Response to Reviewer yHJY (Part 2)**
>
> **6. Regarding tasks other than classification**
>
> CodeTrek does not preclude other kinds of tasks and we will be happy to add a discussion. For instance, setting up a function summarization task is similar to the tasks described in the paper. Since this task requires reasoning over the entire function definition, multiple randomly selected nodes can serve as the anchors. The trained CodeTrek model then computes an embedding of the given function which can be used to generate a natural language summary.
>
>
> **7. The definition of “walk score”**
>
> We thank the reviewer for pointing this out. The walk score is defined in Section 2.3 under “Training and Inference”, and is denoted by the variable $\alpha_w$.
>
>
> **8. Table 3 is interesting. Does it mean that the expected benefit from the deep and often complicated semantic analysis is sometimes as low as 20%? Do you think sampling more paths from the AST could have narrowed the gap?**
>
>
> We assume the reviewer is referring to the Exception-Fun task in Table 3 which shows only a 2% improvement. We did try sampling more paths from the AST (upto 150 paths), but they did not narrow the gap further. This is because there were not many walks that could be sampled solely from the AST.
>
> There is a very insignificant improvement in the performance of the Exception-Fun task because this task primarily relies on the ability of a model to memorize tokens and the code structure to correctly predict the Exception to be caught. Hence, incorporating more semantic information in the graph does not significantly improve the performance. On the other hand, since the Exception task relies on reasoning over called functions within the try block and understanding the exceptions they may throw, incorporating semantic information in the graph resulted in a significant improvement to the performance.

---

### Official Review · Reviewer_n7bP · 2021-11-04

**Correctness:** 4
**Technical Novelty And Significance:** 3
**Empirical Novelty And Significance:** 2
**Recommendation:** 8
**Confidence:** 5

**Main Review:**

I'm both excited and disappointed by this paper. The idea of using relations as obtained by the Semmle tool as intermediate representation is elegant, reasonably extensible, and opens a new path to integrating more knowledge from the expert practicioners into the ML4Code area. I believe that this is a valuable contribution that should be published.

On the other hand, the remainder of the paper has a number of substantial problems:
1. the translation of the database of program facts into a binary graph is not described well:
  * (Q1a) how do tuple attributes exactly contribute to the node features? how are different data types (stmt types / IDs / line spans / ...) embedded? If I understand p5 top correctly, you're simpling bag-of-words embedding WordPieces here - does this even take into account what attribute a token comes from?
  * (Q1b) how fine-grained are edge types? Concretely, when translating something like the "call" relation in Fig. 1, are separate edge types generated for callees and callers? The main text says "for any two relations R and S connected via referential integrity, an R_S edge type is defined". Alg. 2 says "l [...] is a referential integrity constraint", which is an undefined term in the paper.
  * (Q1c) have you considered representing these relations as hypergraphs (with each named relation corresponding to a hyperedge type, and each tuple from a relation instantiation corresponding to a hyperedge) and then using HyperGNN models?

2. the random walk strategy is an interesting trick to break through the limitations of number of message passing steps in GNNs / number of layers in Transformers, but leaves many questions:
  * (Q2a) You say "each individual part of the embedding tensor gets its own learnable positional embedding" - it's not clear to me what that actually means - are you describing an embedding lookup here, or do you mean to say that the relation names / relation attributes / edge types have separate positional encodings? How do you ensure that they are well-aligned with each other? Do you have ablations that show that your positional encodings contribute anything at all?
  * (Q2b) You are using mean pooling to obtain the representation of a path, but that is likely to be sensitive to the length of the path. Have you considered using attentional pooling (e.g., most simple, introducing a [PATH] pseudo-token and use its final representation, or at least a gated, weighted sum)? After all, you're using a similar attentive approach for the aggregation of paths by going through DeepSet.

3. the experiment section feels not well-structured (some explicit guideposting what key points you are trying to support would help with this), and the considered baselines are underdescribed and seem incomplete.
  * (Q3a) The hypers described in App. A for the baselines are surprising in that they don't match the hyperparameters proposed by the original authors - in particular, GREAT (4 layers here vs. 6 or 10 layers), Code2Seq (4 LSTM layers here vs. 1 in original), and GGNN (4 layers here vs. 8 layers). How where these parameters chosen? In particular, the low number of layers for GREAT and GGNN substantially limits their performance, and so this is quite surprising.
  * (Q3b) What edges are presented to the GREAT / GGNN models, which consume more than just the AST? This is not described at all.
  * (Q3c) How sensitive are your results to different choices of the random walk specification? You list learning the spec as future work, but what is the effect of choices here? How did you arrive at your current default spec?



**Summary Of The Paper:**

A method to learn to reason about programs is presented. The core novelty is an intermediate program representation as a set of relations (in the logics sense), derived from existing program analysis tools. By translating each tuple into a node, and references between relations into edges, the set of relations is transformed into a graph. Finally, graph learning techniques (from the random walk view of the world) are used to learn to predict the target label from the obtained graphs.

**Summary Of The Review:**

The paper presents a novel way of automatically extracting valuable information from programs before using standard ML tools to process them. While the core idea is exciting, the writing is in some places imprecise (see above) and the experimental evaluation is missing many details and leaves many question marks on the empirical claims. I like this paper, but I think at this time, it's shortcomings are too significant to allow for acceptance at ICLR.


Update after rebuttal
----------------------------
The authors did a great job of responding to my concerns and answering my questions, alleviating most of my concerns.

I would note that the current draft does not integrate the further explanations and results responding to my question (Q1a).

The draft also has not been updated to include the additional experimental results and ablations that the authors provided during the discussion period, and I would encourage them to include all of these in the next revision of the paper, either embedded into the main text or in an appendix.

Overall, I'm now satisfied that the paper is presenting an empirically meaningful improvement over earlier works. As the reviewing discussion highlighted, constructing program graphs as proposed by Allamanis et al. (and following works based on the extraction of semantic program information) is highly non-trivial and not an off-the-shelf product. I view the core contribution of the current submission as the empirically validated insight that this can be solved by the Semmle static analysis tool. While many open questions about design choices here remain, this is a substantial step forward that deserves publication at ICLR, as I expect it to be of significant interest to the ML4Code subcommunity at the conference.

---

> ### Author Response · Authors · 2021-11-15
> **Author Response to Reviewer n7bP (Part 1)**
>
> **1.**
>
> We have updated Section 2 and divided Figure 1 into three figures to clarify the main components of CodeTrek.  The new Figure 1 shows how the Semmle compiler represents the input Python program as a relational database.  Figure 2 shows a walk in the graph that CodeTrek constructs from the database.  Figure 3 shows the embedding of the walk.
>
> **Q1(a):**
>
> As Figure 3 illustrates, we embed all literals besides key/foreign-key values using a subtoken vocabulary (e.g. the `"except"` stmt kind, the `“test.py:130-138”` line span, etc. but not stmt ID `“s1”` or scope ID `“c1”`). We did not take into account what attribute a token comes from because relations tend to have a few such attributes (most relations have just one), and the contents of the literals are more informative than which attributes they come from.
>
> **Q1(b):**
>
> Using separate edge types `“func_call”` and `“call_func”` for callers and callees does not perform better than using the same edge type.  More generally, suppose R.A (attribute A in relation R) references S.B (attribute B in relation S).  Then, we define a single edge type R_S.  For example, since the `“call”` relation has two attributes `“caller”` and `“callee”` each referencing attribute `“id”` in the `“func”` relation, we define the same edge type `“call_func”`.
>
> We provided the formal definition of referential integrity constraint in Appendix C.2.  We have now also updated Figure 1 with an example of such a constraint and explained it in Section 2.1 (Background).
>
> **Q1(c):**
>
> This is a very interesting question.  We will qualitatively compare our graph representation to that in HyperGNN models. The key difference between a hypergraph and our proposed graph is the kind of information that each node carries. In a hypergraph, any occurence of a specific literal or key is represented as a node. This means that all the hypernodes that have a literal in common in their corresponding tuples, are connected. The challenge that this hypergraph causes is that it makes connections between tuples which are not semantically meaningful. For example, tuples `number(ID: n1, Value: “0”)` and `number(ID: n101, Value: “0”)` that represent two numbers in a program would be connected through the node that represents the `“0”` literal, although there are no semantic dependencies between these two numbers. As a result, the sampled random walks are likely to capture meaningless dependencies, possibly hampering the performance of the model. Additionally, HyperGNNs embed the node contents, which may include tokens and any associated attributes like node types, subtokens, etc., and only consider the graph structure during message passing. On the other hand, CodeTrek samples walks from the graph and embeds node contents as well as the structure represented by the edge types with the same weight. This also allows us to incorporate the node contents and structure more integrally into the CodeTrek models rather than using the node contents and their attributes once and then solely focusing on the structure, as one would in HyperGNN models.
>
> **2.**
>
> **Q2(a):**
>
> We have added Figures 2 and 3 to clarify these points. We have also added clarification in the Neural Encoder paragraph in Section 2.3. To embed a walk, we first break it into three separate sequences: 1) the sequence of node types, i.e. the relation names that the walk tuples correspond to, 2) the sequence of node values which are the literals in the walk tuples, and 3) the sequence of edge types. Each of these sequences are first embedded using node vocabulary, subtoken vocabulary, and edge type vocabulary, respectively, and each has a separate positional encoding.
>
> We only need to ensure that within each one of these three parts, the position encoding captures the ordering of walks. Whether the embeddings of a given position in different parts equal to each other does not really matter -- and actually this brings a bit more flexibility, which lets the model understand the roles of different parts.
>
> The reviewer raises an interesting point regarding the contribution of the positional encodings. We are currently running ablations for the positional encoding and will report the results as soon as they are ready.
>
>
> **(Q2b):**
>
> The reviewer raises an interesting point regarding our choice for the pooling mechanism. We are running experiments to 1) measure the sensitivity of the model to different lengths of the path when using mean poolings, and 2) measure the performance of the model when using attention pooling instead of mean pooling. We will report the results as soon as they are ready.

---

> ### Author Response · Authors · 2021-11-15
> **Author Response to Reviewer n7bP (Part 2)**
>
> **3.**
>
> **(Q3a):**
>
> Our focus during the experiments was primarily on the various representations used by the different baselines. In order to reflect the effectiveness of these representations on their respective models, we chose to keep the hyperparameters consistent across all the baselines wherever possible.
>
> **(Q3b):**
>
> The edges that are presented to GGNN models are borrowed from Allamanis et al. (ICLR’18). They include AST edges, NextToken edges, LastRead/LastWrite/ComputedFrom/LastLexicalUse edges among variable accesses, GuardedBy/GuardedByNegation edges between variables used in branches and their corresponding conditional expressions, ReturnsTo edges from the return statement to the method declaration, and FormalArgName edges between method call arguments and their corresponding formal parameters. For GREAT models, similar to Hellendoorn et al. (ICLR’20), we borrow the edge types from Allamanis et al. (ICLR’18) and augment them with function calls. We have updated Appendix F to include these details.
>
> **(Q3c):**
>
> _Number of walks:_ We have experimented with choices for the number of random walks. We found that performance generally improves with more walks upto a certain point (100 walks) but does not improve noticeably when the number of walks is over 100. So we picked this value as the default number of walks.
>
> _Relations:_ We include only the base relations in the default specification. The only special case is the Exception task, where we further include a derived relation `“call”`---which relates functions with their callers---which improves the accuracy of this task from 52% to 63%.
>
> _Biases:_ The primary purpose of biases over nodes is to guide the walk generator to sample more relevant walks by reducing the probability of traversing nodes that may diverge from the relevant portions of the graph. For example, if a walk was to traverse to its parent `“module”` node, it may then traverse to other child nodes within the module that may not be relevant to the problem. On the other hand, node types like `“expr”`, `“stmt”`, and `“variable”` tend to be more relevant while also relating to other useful nodes. Hence we give these node types a higher bias in the default specification than other node types. We do examine the effect of the absence of these biases on the Exception-Fun task in Section 3.4.

---

> > ### Comment · Reviewer_n7bP · 2021-11-18
> > **Rebuttal Discussion**
> >
> > Thank you for your detailed answers to my questions. I have three follow-ups to this:
> >
> > > Q1(b):
> > >
> > > Using separate edge types “func_call” and “call_func” for callers and callees does not perform better than using the same edge type. More generally, suppose R.A (attribute A in relation R) references S.B (attribute B in relation S). Then, we define a single edge type R_S. For example, since the “call” relation has two attributes “caller” and “callee” each referencing attribute “id” in the “func” relation, we define the same edge type “call_func”.
> >
> > This is a very surprising result to me - shouldn't it matter for the model whether it is looking at a callee vs. a caller? Do you have any intuition of why this does not impact performance?
> >
> > > Q2(a):
> > >
> > > [...] Each of these sequences are first embedded using node vocabulary, subtoken vocabulary, and edge type vocabulary, respectively, and each has a separate positional encoding.
> >
> > Just to confirm, the positional encodings are learned (i.e., not sinusoidal) and have separate parameters for the different parts of a walk, correct? In that case, I'm confused as to how the model is supposed to learn how to align the three parts of your walk encoding, and in particular, I doubt that it is able to learn the connections between different nodes in the walk (which would explain why the callee/caller differentiation doesn't matter).
> > Have you experimented with alternative schemes? One natural scheme would be to interleave the three sub-tensors and to use a single (potentially parameter-free sinusoidal) positional encoding. This should be easy to do by padding the `edge_types` value with a dummy value at the end, stacking the three individually embedded tensors, and then reshaping it into single long sequence.
> >
> > > (Q3a):
> > >
> > > Our focus during the experiments was primarily on the various representations used by the different baselines. In order to reflect the effectiveness of these representations on their respective models, we chose to keep the hyperparameters consistent across all the baselines wherever possible.
> >
> > What do you mean by consistent here? Consistent in that you use 4 everywhere?
> >
> > I'm very concerned by these choices, as it, in my eyes, entirely invalidates the experimental comparison. You are comparing your models, able to do walks of length 24 on your graph, with models that can at most consider 4 steps in the graph (in the case of the GNN and GREAT baselines). Especially given the fact that these are substantially smaller than the author-provided recommendations, this makes it look like you are intentionally limiting the performance of your baselines, which seems very concerning to me.

---

> > > ### Author Response · Authors · 2021-11-22
> > > **Author Response to Reviewer n7bP**
> > >
> > > > **This is a very surprising result to me - shouldn't it matter for the model whether it is looking at a callee vs. a caller?**
> > >
> > > We apologize for the confusion that our response caused. As illustrated in Figure 1 of the revised draft, the relations involved in caller/callee relationships are `func(id, name, scope)` and `call(id, caller, callee)`. The possible edges between these relations exist between `func.id` and `call.caller`, and between `func.id` and `call.callee`. Therefore we consider _two_ types of edges between relations `func` and `call`, one being `“func_call”` between `func.id` and `call.caller`, and the other being `“func_call”` but between `func.id` and `call.callee`.
> > >
> > > What we do not do, however, is add edge types `“call_func”` for each of the aforementioned edge types. Concretely, since `“func_call”` exists between `func.id` and `call.caller`, we will not add another edge type `“call_func”` between `func.id` and `call.caller`, since both are essentially the same edge type but with different names.
> > >
> > >
> > > **Clarification regarding the positional encoding schema:**
> > >
> > > We have implemented both schemes in our early experiments:
> > > - Using learned position encoding for each one of the three parts (node-value, node-type, edge-type) as we stated in the paper.
> > > - Using the same sinusoidal position encoding for each part (each part starts the position counting from 1). This should be equivalent to what you suggested.
> > > We want to make a correction on the results we reported before. Using the learned position gives us **64.65** accuracy v.s. the **63.68** with sinusoidal encoding. Previously we reported 63.68 in the paper due to the mistake in the result collection. We apologize for not checking it carefully and we thank the reviewer for being rigorous on the modeling details. Overall we think that the two design choices should be somehow similar, as what is pointed out in the [vanilla Transformer paper](https://arxiv.org/abs/1706.03762): *“We also experimented with using learned positional embeddings …, the two versions produced nearly identical results”*.
> > >
> > > > **I'm confused as to how the model is supposed to learn how to align the three parts of your walk encoding...**
> > >
> > > One design choice is to share the three learned embeddings with the same embedding matrix. In this way it achieves the *‘alignment’* by design. Untying these three embeddings is a strict generalization in the model capacity (where the tied version is a special case) and offers more freedom for the modeling. In fact we are not sure whether we have to *align* these parts, as the Transformer attention works in an order-less way and what indeed matters are the pairwise attention scores (e.g., the relative position encoding [Dai et.al](https://arxiv.org/abs/1901.02860)). Having the *aligned* position encoding implicitly assumes that position 1 in part 1 should have higher affinity to position 1 in part 2 or 3 (due to the cosine similarity is 1 between the two positional encoding, and the additive association of position encoding and token embedding), which might be a good inductive bias but might not always be true. So instead we let the model learn what adapts best.
> > >
> > > > **... I'm very concerned by these choices, as it, in my eyes, entirely invalidates the experimental comparison ...**
> > >
> > > To address your concern, we have re-run GREAT and GGNN, each with 10 layers instead of 4,  for every task for which CodeTrek outperforms the baseline. We report the performance results in the table below.
> > >
> > > |Task        |CodeTrek|GGNN(10L)|GREAT(10L)|
> > > |:----------:|:------:|:-------:|:--------:|
> > > |Varmisuse   |90.5    |71.5     |??        |
> > > |VarmisuseFun|70.0    |55.2    |> CodeTrek|
> > > |Exception   |63.0    |30.1    |44.7      |
> > > |ExceptionFun|65.1    |53.3     |> CodeTrek|
> > > |Defuse      |97.7    |78.0       |??        |
> > > |DefuseFun   |91.0    |77.7     |84.8      |
> > > |VarShadow   |93.8    |73.6     |93.5      |
> > >
> > > For GGNN models with 10 layers, we observed 0.7% to 3% points increase in their accuracy. CodeTrek still outperforms these GGNN models. For GREAT models with 10 layers, so far, we have obtained the results for Exception, DefuseFun, and VarShadow. Compared to the corresponding GREAT models with 4 layers, these models show 0.8%, 2.5%, and 0.7% points increase in the accuracy, respectively. According to these new results, the gap between the performance of GREAT models (with 10 layers) and CodeTrek is slightly smaller than the performance of GREAT models (with 4 layers) and CodeTrek. As reported in the table, VarShadow has almost the same accuracy in both models (93.5 in GREAT vs. 93.8 in CodeTrek).
> > >
> > > As noted by the original authors of the paper that introduced GREAT, the training of these larger models can take up to 240 hours on one Tesla P100 GPU. Due to this long training time, we are still training GREAT models with 10 layers for Varmisuse and Defuse and will report the results as soon as they are ready.

---

> > > > ### Comment · Reviewer_n7bP · 2021-11-22
> > > > **Discussion**
> > > >
> > > > >> This is a very surprising result to me - shouldn't it matter for the model whether it is looking at a callee vs. a caller?
> > > >
> > > > > We apologize for the confusion that our response caused. As illustrated in Figure 1 of the revised draft, the relations involved in caller/callee relationships are func(id, name, scope) and call(id, caller, callee). The possible edges between these relations exist between func.id and call.caller, and between func.id and call.callee. Therefore we consider two types of edges between relations func and call, one being “func_call” between func.id and call.caller, and the other being “func_call” but between func.id and call.callee.
> > > >
> > > > I'm pleased to hear that these are distinguished, but I have to admit I still don't quite understand how a foreign key definition in the database schema is translated into an edge name. Maybe you could provide an exact definition in the paper in your next revision?
> > > >
> > > > >> I'm confused as to how the model is supposed to learn how to align the three parts of your walk encoding...
> > > >
> > > > > [...] In fact we are not sure whether we have to align these parts, as the Transformer attention works in an order-less way and what indeed matters are the pairwise attention scores (e.g., the relative position encoding Dai et.al). Having the aligned position encoding implicitly assumes that position 1 in part 1 should have higher affinity to position 1 in part 2 or 3 (due to the cosine similarity is 1 between the two positional encoding, and the additive association of position encoding and token embedding), which might be a good inductive bias but might not always be true. So instead we let the model learn what adapts best.
> > > >
> > > > My counterpoint here is that given your current scheme, the model _needs to learn_ which edges connect which nodes. By introducing positional encoding per "part", you are providing an order on each part, but it remains unclear if the model learns to align these with each other. I firmly believe that this is a part of the model in which changes are especially promising.
> > > >
> > > > > To address your concern, we have re-run GREAT and GGNN, each with 10 layers instead of 4, for every task for which CodeTrek outperforms the baseline.
> > > >
> > > > Thank you, these updated experiments convince me to a much higher degree that your system is indeed working better than existing baselines in practice.
> > > >
> > > > While there are certainly still open issues (w.r.t. clarity, alignment with conference audience, etc.), this makes me much more positive about the paper, and I'm happy to raise my score. I'll do so once the discussion phase ends and have reviewed all your answers in one go.

---

> > > > > ### Author Response · Authors · 2021-11-22
> > > > > **Author Response to Reviewer n7bP**
> > > > >
> > > > > We thank the reviewer for their insightful and constructive comments. We will incorporate all the feedback to improve the clarity of the paper.

---

> ### Author Response · Authors · 2021-11-17
> **Author Response to Reviewer n7bP (Part 3)**
>
> > **(Q2a) Do you have ablations that show that your positional encodings contribute anything at all?**
>
> Our ablation study on the effect of the positional encoding shows that it contributes 1.77% points to the accuracy of the Exception task, i.e., by skipping the positional encoding, the accuracy drops from 63.83% to 62.06% points.
>
> > **(Q2b) Have you considered using attentional pooling?**
>
> We did an experiment to compare the accuracy of CodeTrek on the Exception task with the exact same configuration but with different pooling mechanisms. When we use mean pooling, we obtain an accuracy of 63.83% points. However, using attentional pooling mechanism increases the accuracy to 66.43% points.

---

### Author Response · Authors · 2021-11-22
**Official Comment**

We thank all the reviewers for their constructive feedback. In both the previous and the new revision, we have tried to address the comments regarding the clarity. We have marked the text that we changed in this revision by red font:

- On page 3, we added a more clear definition of the referential integrity at the end of Section 2.1, updated Figure 2 to show the complete edge types between `"call"` and `"func"`, and explained how key-foreign-key relationships are mapped to edge types in Section 2.2.
- On page 5, we updated the last paragraph of "Code as a Graph" for clarification.

---

### Author Response · Authors · 2021-11-27
**Official Comment**

Dear reviewers, since the author discussion period is ending soon, we are wondering if there are any remaining questions or concerns we can address. We will incorporate all the feedback we have received from you. We thank you all for your time.

---

### Decision · Program_Chairs · 2022-01-20

**Decision:**

Accept (Poster)

**Comment:**

The paper presents a deep learning approach encodes codebases as databases that conform to rich relational schemas. Based on this, a biased graph-walk mechanism efficiently feeds this structured data into a transformer and deepset approach. The results shown a quite good, compared to other approaches present at ICLR. Moreover, one reviewer is strongly voting for accepting the paper, arguing that "that this paper is of significance to the ML4Code research community, as it shows how to offload the engineering cost of extracting semantic information from programs to a standard tool." Overall, I have really enjoyed reading the paper, and the use of relational database as codebase together with a transformers is sweat. On the other, it also presented in a rather engineering way, as pointed out by several reviewers, suggesting that some software engineering venue might be a better place for the work. But then ICLR had similar papers, and the present paper demonstrates a benefit of using a relational encoding. Thus, I weight the leaning towards rejects borderlines votes less and suggest an accept overall. We all should keep in mind that also deep neural architecture are full of design choices.